# Inter-tissue convergence of gene expression during ageing suggests age-related loss of tissue and cellular identity

**Hamit Izgi[1], Dingding Han[2†], Ulas Isildak[1], Shuyun Huang[2], Ece Kocabiyik[1], Philipp Khaitovich[3]\*, Mehmet Somel[1]\*, Handan Melike Dönertaş[4,5]\***

[1]Department of Biological Sciences, Middle East Technical University, Ankara, Turkey; [2]CAS Key Laboratory of Computational Biology, CAS-MPG Partner Institute for Computational Biology, Shanghai Institutes for Biological Sciences, Chinese Academy of Sciences, Shanghai, China; [3]Center for Neurobiology and Brain Restoration, Skolkovo Institute of Science and Technology, Moscow, Russian Federation; [4]European Molecular Biology Laboratory, European Bioinformatics Institute EMBL-EBI, Wellcome Trust Genome Campus, Cambridge, United Kingdom; [5]Leibniz Institute on Aging - Fritz Lipmann Institute (FLI), Jena, Germany

**\*For correspondence:**
p.khaitovich@skoltech.ru (PK);
somel.mehmet@googlemail.com (MS);
melike.donertas@leibniz-fli.de (HMD)

**Present address:** †Department of Clinical Laboratory, Shanghai Children's Hospital, Shanghai Jiaotong University, Shanghai, China

**Competing interest:** The authors declare that no competing interests exist.

**Abstract** Developmental trajectories of gene expression may reverse in their direction during ageing, a phenomenon previously linked to cellular identity loss. Our analysis of cerebral cortex, lung, liver, and muscle transcriptomes of 16 mice, covering development and ageing intervals, revealed widespread but tissue-specific ageing-associated expression reversals. Cumulatively, these reversals create a unique phenomenon: mammalian tissue transcriptomes diverge from each other during postnatal development, but during ageing, they tend to converge towards similar expression levels, a process we term *Di*vergence followed by *Co*nvergence (DiCo). We found that DiCo was most prevalent among tissue-specific genes and associated with loss of tissue identity, which is confirmed using data from independent mouse and human datasets. Further, using publicly available single-cell transcriptome data, we showed that DiCo could be driven both by alterations in tissue cell-type composition and also by cell-autonomous expression changes within particular cell types.

## Editor's evaluation

In this study, Izgi et al. investigated age-dependent gene expression pattern changes in male mice by analysing a new bulk RNA-seq data from four different tissues collected at different ages covering postnatal development and ageing. Gene expression patterns observed before sexual maturity show inter-tissue divergence, whereas convergence of gene expression profiles is observed after sexual maturity and during ageing, in a pattern that the authors call divergence-convergence or 'DiCo.' This observation may suggest that ageing results in at least a partial loss of tissue identity acquired developmentally.

## Introduction

Development and ageing in multicellular organisms are highly intertwined processes. On the one hand, certain ageing-related phenotypes, such as presbyopia and osteoporosis (*Luegmayr et al., 2004*), are believed to represent the continuation of developmental processes into adulthood (*Blagosklonny, 2006*; *de Magalhães and Church, 2005*). Such cases of 'runaway development' or higher than optimal function during ageing (recognised as the hyperfunction theory of ageing; *Gems and*

*Partridge, 2013*) may arise due to declined natural selection pressure failing to optimise expression regulation after sexual reproduction starts (*Fisher, 1930*; *Medawar, 1953*; *Williams, 1957*). Indeed, recent experimental studies in *Caenorhabditis elegans* show that senescence phenotypes promoted by insulin-IGF-1 signalling pathways support the hyperfunction theory (*Lind et al., 2019*; *Ezcurra et al., 2018*). On the other hand, molecular studies have also reported a reversal of the ageing transcriptome towards pre-adult levels in various contexts, including primate brain regions (*Somel et al., 2010*; *Dönertaş et al., 2017*; *Colantuoni et al., 2011*), and mouse liver and kidney (*Anisimova et al., 2020*). Studying the functional consequences of this reversal pattern in the ageing human brain, we previously interpreted it as an indication of loss of cellular identity in neurons, possibly exacerbated by a reduction in the relative frequencies of neurons (*Dönertaş et al., 2017*). Such changes, in turn, could be caused by the accumulation of stochastic damage at the genetic, epigenetic, and proteomic levels over an adult lifetime, causing deregulation of gene expression networks.

Several major questions remain. First, the prevalence of reversal phenotypes across tissues is unclear as most research has been conducted in the brain (*Somel et al., 2010*; *Dönertaş et al., 2017*). A second question pertains to the similarity of reversal-exhibiting genes and pathways across tissues. Ageing-related expression changes are partly shared among organs (*Zahn et al., 2007*), and reversal trends are also shared across different regions of the primate brain (*Dönertaş et al., 2017*). Distinct tissues might hence show parallel reversal patterns. Alternatively, as mammalian tissues diverge from each other during development in their transcriptome profiles (*Cardoso-Moreira et al., 2019*), one may hypothesise that during ageing tissues converge back towards similar transcriptome profiles. Such a putative late-age convergence phenomenon would be consistent with the notion of ageing-related cellular identity loss (*Yang et al., 2019*; *Dönertaş et al., 2017*). A final question concerns the mechanism behind the observed reversal trends at the bulk tissue level. Specifically, the contribution of cell-type composition and cell-autonomous changes to the reversals at the tissue level remains unexplored.

Documenting the reversal phenomenon is critical to better understand the proximate mechanisms of mammalian ageing, and its ultimate mechanisms, such as the stochastic disruption versus continued expression of developmental genes. However, such work has been limited by the scarcity of studies that include both development and ageing periods of the same organism and across different tissues. This work presents an age-series analysis of bulk transcriptome profiles of mice, including samples of four tissues across postnatal development and ageing periods covering the whole postnatal lifespan. Using this dataset, we study the prevalence, mechanisms, and functional consequences of the reversal phenomenon in different mouse tissues. We further test the related hypothesis of tissue convergence during ageing and investigate the contribution of cell-type composition and cell-autonomous changes.

## Results

We generated bulk RNA-seq data from 63 samples covering the cerebral cortex (which we refer to as cortex), liver, lung, and skeletal muscle (which we refer to as muscle) of 16 male C57BL/6J mice, aged between 2 and 904 days of postnatal age (Materials and methods). As mice reach sexual maturity by around 2 months (*Tacutu et al., 2018*), we treated samples from individuals aged between 2 and 61 days (n = 7) as the development series, and those aged between 93 and 904 days (which roughly correspond to 80-year-old humans; *Flurkey et al., 2007*) (n = 9) as the ageing series (*Figure 1—figure supplement 1*). The final dataset contained n = 15,063 protein-coding genes expressed in at least 25% of the 63 samples (one 904-day-old mouse lacked cortex data).

### Tissues diverge during postnatal development

Consistent with earlier work (*Brawand et al., 2011*; *Cardoso-Moreira et al., 2019*), we found that variation in gene expression is largely explained by tissue differences, such that the first three principal components (PCs) separate samples according to tissue (ANOVA p<10$^{-20}$ for PC1–3, *Figure 1—source data 1*), with the cortex most distant from the others (*Figure 1a*). Meanwhile, PC4, which explains 8% of the total variance, displayed a shared age effect across tissues in development (Spearman's correlation coefficient $\rho$ = [-0.88, –0.99], nominal p<0.01 for each test; *Figure 1b*). Also, after the tissue effect was removed by standardisation, principal components analysis (PCA) showed a strong influence of age on the first two PCs, which explains 31% of the variance in total (*Figure 1—figure*

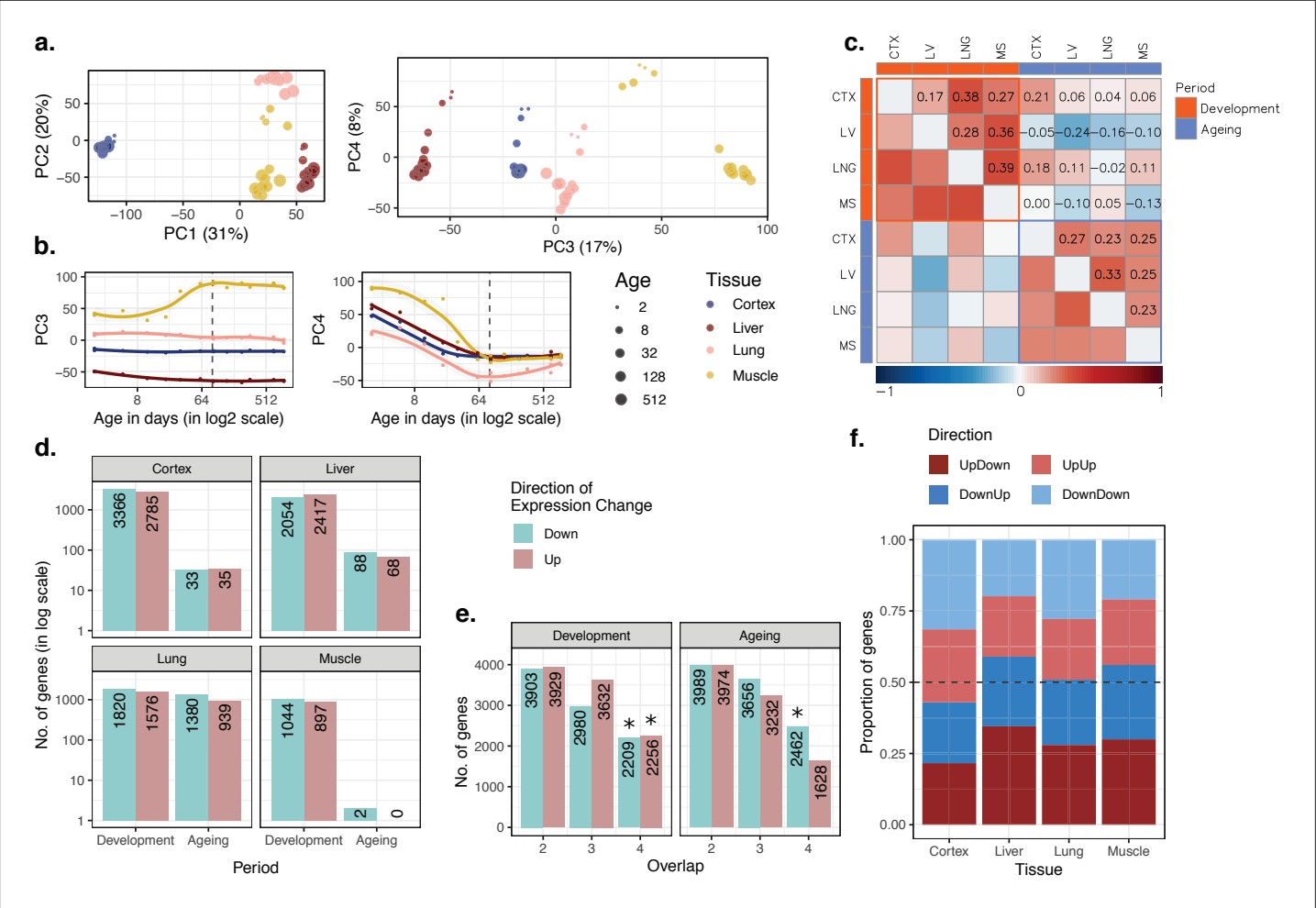

**Figure 1.** Data summary and age-related expression patterns. (**a**) Principal components analysis (PCA) of expression levels of 15,063 protein-coding genes across four tissues of 16 mice. Values in parentheses show the variation explained by each component. (**b**) Age trajectories of PC3 (left) and PC4 (right). Spearman's correlation coefficients between PC4 and age in each tissue in development range between 0.88 and 0.99 (see *Figure 1—source data 1* for all tests). The dashed vertical line indicates 90 days of age, separating development and ageing periods. Age distribution of samples are given in *Figure 1—figure supplement 1*. (**c**) Similarity between the age-related gene expression changes (Spearman's correlation coefficient between expression and age without a significance cutoff) across tissues in development and ageing. Similarities were calculated using Spearman's correlation coefficient between expression-age correlations across tissues. CTX, cortex; LV, liver; LNG, lung; MS, muscle. (**d**) The number of significant age-related genes in each tissue (false discovery rate [FDR]-corrected p-value<0.1). (**e**) Shared age-related genes among tissues identified without using a significance cutoff. The x-axis shows the number of tissues among which age-related genes are shared. Significant overlaps are indicated with an asterisk (*Figure 1—figure supplement 4*). (**f**) The proportion of age-related expression change trends (no significance cutoff was used) in each tissue across the lifetime. UpDown: upregulation in development and downregulation in the ageing; DownUp: downregulation in development and upregulation in the ageing; UpUp: upregulation in development and upregulation in the ageing; DownDown: downregulation in development and downregulation in the ageing. We confirmed the robustness of the results using variance stabilising transformation (VST normalisation in *Figure 1—figure supplement 10*).

The online version of this article includes the following source data and figure supplement(s) for figure 1:

**Source data 1.** Data summary, age-related expression patterns, and reversal patterns.

**Figure supplement 1.** Age distribution of samples.

**Figure supplement 2.** Principal components analysis (PCA) with all samples (tissue effect removed).

**Figure supplement 3.** Principal components analysis (PCA) with development and ageing periods separately.

**Figure supplement 4.** Permutation test results for shared expression trends among tissues.

**Figure supplement 5.** Shared age-related genes among tissues in development and ageing.

**Figure supplement 6.** Permutation test results for significant trends shared among tissues.

**Figure supplement 7.** Similarities between age-related gene expression changes among tissues.

*Figure 1 continued on next page*

*Figure 1 continued*

**Figure supplement 8.** Permutation test results for reversal patterns in each tissue.

**Figure supplement 9.** Permutation test results for shared reversals among tissues.

**Figure supplement 10.** Replication of *Figure 1* results using variance stabilising transformation (VST) normalisation.

**Figure supplement 11.** Correlation between quantile normalised (QN) and variance stabilising transformation (VST) normalisation methods using age-related expression changes.

**Figure supplement 12.** Clustering of genes by expression levels in cortex tissue.

**Figure supplement 13.** Clustering of genes by expression levels in lung tissue.

**Figure supplement 14.** Clustering of genes by expression levels in liver tissue.

**Figure supplement 15.** Clustering of genes by expression levels in muscle tissue.

*supplement 2*). We further observed higher similarity among tissues at the juvenile stage compared to the young-adult stage. In other words, distances between tissues increased with age (change in mean Euclidean distance among tissues with age during development in PC1–PC4 space $\rho_{dev} = 0.99$, $p_{dev} = 1.5 \times 10^{-5}$, *Figure 1—source data 1*), which resonates with previous reports of inter-tissue transcriptome divergence during development (*Cardoso-Moreira et al., 2019*). This divergence pattern was also observed when PCA was performed with developmental samples only (days 2–61: change in mean Euclidean distance among tissues in PC1–PC4 space; $\rho = 0.95$, p=0.0008; *Figure 1—figure supplement 3a and b*).

## Tissues involve common gene expression changes with age

We next characterised age-related changes in gene expression shared across tissues by (1) studying overall trends at the whole transcriptome level and testing their consistency using permutation tests, and (2) studying statistically significant changes at the single-gene level. First, we investigated similarities in overall trends of gene expression changes with age using the Spearman's correlation coefficient ($\rho$) between expression levels and age for each gene in each tissue separately for the developmental and ageing periods (Materials and methods; tissue-specific age-related gene expression changes and functional enrichment test results are available in *Supplementary file 1*). We then examined transcriptome-wide similarities across tissues during development and ageing by comparing these gene-wise expression-age correlation coefficients (*Figure 1c*). Considering the whole transcriptome without a significance cutoff, we found a weak correlation of age-related expression changes in tissue pairs, both during development ($\rho = [0.17, 0.39]$, permutation test p<0.05 for all the pairs, *Figure 1—source data 1*), and ageing ($\rho = [0.23, 0.33]$, permutation test p<0.05 in 4/6 pairs, *Figure 1—source data 1*). We then tested whether developmental patterns among tissues may be shared more than ageing-associated patterns, but we did not find significant difference between inter-tissue similarities within the development and those within ageing (Wilcoxon signed-rank test, p=0.31). Moreover, the number of genes with the same direction of change (without applying a significance cutoff) across four tissues was consistently more than expected by chance (permutation test p<0.05), except for genes upregulated in ageing (*Figure 1e*, *Figure 1—figure supplement 4*). This attests to overall similarities across tissues both during postnatal development and during ageing, albeit of modest magnitudes. We obtained similar results using another normalisation approach, variance stabilising transformation (VST) from the DESeq2 package (*Love et al., 2014*), and confirmed that the observed patterns are not affected by the choice of normalisation method (*Figure 1—figure supplements 10–11*).

In the second approach, we focused on genes showing a significant age-related expression change, identified separately during development or during ageing (using Spearman's correlation coefficient and false discovery rate [FDR]-corrected p-value<0.1, *Figure 1d*). We found that the developmental period was accompanied by a large number of significant changes (n = [1,941, 6,151], 13–41% across tissues), with the most manifest changes detected in the cortex. The genes displaying significant developmental changes across all four tissues also showed significant overlap (*Figure 1—figure supplement 5a*, *Figure 1—figure supplement 6*; permutation test: $p_{shared\_up} = 0.027$, $p_{shared\_down} < 0.001$). Using the Gene Ontology (GO), we found that shared developmentally upregulated genes were enriched in functions such as hormone signalling pathways and lipid metabolism (FDR-corrected p-value<0.1). Meanwhile, shared developmentally downregulated genes were enriched in functions such as cell cycle and cell division (FDR-corrected p-value<0.1; *Supplementary file 2*). Contrary to

widespread expression change during development (13–41%), the proportion of genes undergoing significant expression change during ageing was between 0.013 and 15% (*Figure 1d*). This contrast between postnatal development and ageing was also observed in previous work on the primate brain (*Somel et al., 2010*; *Işıldak et al., 2020*). In terms of the number of genes with a significant ageing-related change, the most substantial effect we found was in the lung (n = 2319), while close to no genes showed a statistically significant change in the muscle (n = 2), a tissue previously noted for displaying a weak ageing transcriptome signature across multiple datasets (*Turan et al., 2019*). Not unexpectedly, we found no common significant ageing-related genes across tissues (*Figure 1—figure supplement 5a*). Considering the similarity between the ageing and development datasets (*Figure 1c*) and the similar sample sizes in development (n = 7) and ageing periods (n = 9), the lack of overlap in significant genes in ageing might be due to low signal-to-noise ratios in the ageing transcriptome as ageing-related changes are subtler compared to those in development (*Figure 1—figure supplement 5b*).

## Gene expression reversal is a common phenomenon in multiple tissues

We then turned to investigate the prevalence of the reversal phenomenon (i.e. an opposite direction of change during development and ageing) across the four tissues. We first compared the trends of age-related expression changes between development and ageing periods in the same tissue, without a significance cutoff, to assess transcriptome-wide reversal patterns (*Figure 1c*). This revealed weak negative correlation trends in liver and muscle (though not in the lung and cortex), that is, genes up- or downregulated during development tended to be down- or upregulated during ageing, respectively. These reversal trends were comparable when the analysis was repeated with the genes showing relatively high levels of age-related expression change ($|\rho| > 0.6$ in both periods; *Figure 1—figure supplement 7*). We further studied the reversal phenomenon by classifying each gene expressed per tissue (n = 15,063) into those showing up- or downregulation during development and during ageing. Here, again, we did not use a statistically significance cutoff and summarised trends of continuous change versus reversal in each tissue. This approach follows *Dönertaş et al., 2017* and focuses on global trends instead of single genes. In line with the above results, as well as earlier observations in the brain, kidney, and liver (*Dönertaş et al., 2017*; *Anisimova et al., 2020*), we found that ~50% (43–58%) of expressed genes showed reversal trends (*Figure 1f*), although these proportions were not significantly more than randomly expected in permutation tests (*Figure 1—figure supplement 8*, Materials and methods). Overall, we conclude that although the reversal pattern is not ubiquitous, the expression trajectories of the genes do not necessarily continue linearly into the ageing period.

## Pathways related to development, metabolism, and inflammation are associated with the reversal pattern

We then asked whether genes displaying reversal patterns in each tissue may be enriched in functional categories. Our earlier study focusing on different brain regions had revealed that up-down genes, that is, genes showing developmental upregulation followed by downregulation during ageing, were enriched in tissue-specific pathways, such as neuronal functions (*Dönertaş et al., 2017*). Analysing up-down genes compared to all genes upregulated during development, we also found significant enrichment (FDR-corrected p-value<0.1) in functions such as 'synaptic signalling' in the cortex, as well as 'tube development' and 'tissue morphogenesis' in the lung, 'protein catabolic process' in the liver, and 'cellular respiration' pathways in the muscle (*Supplementary file 3*). Meanwhile, down-up genes (downregulation during development followed by upregulation during ageing) showed significant enrichment in functions such as 'wound healing' and 'peptide metabolic process' in the cortex, 'translation' and 'nucleotide metabolic process' in the lung, 'inflammatory response' in the liver, and 'leukocyte activation' in the muscle (*Supplementary file 3*).

## Genes showing a reversal pattern are not shared among tissues

As tissues displayed modest positive correlations in their development- or ageing-related expression change trends (*Figure 1c*, *Figure 1—figure supplement 7*), and as we had previously observed that distinct brain regions show similarities in their reversal patterns (i.e. the same genes showing the same reversal type), different tissues might also be expected to show similarities in their reversal patterns. Interestingly, we found no overlap between gene sets with the reversal pattern (up-down or down-up genes) across tissues, relative to random expectation (permutation test, $p_{up-down} = 0.08$, $p_{down-up} = 0.53$;

*Figure 1—figure supplement 9*). Such a lack of overlap might be explained if genes showing reversal patterns in each tissue tend to be tissue-specific. It would also be consistent with the notion that reversals involve loss of cellular identities gained in development, during which tissue transcriptomes appear to diverge from each other (*Figure 1a*, *Figure 1—figure supplement 3*; *Cardoso-Moreira et al., 2019*). This result led us to ask whether, in accordance with the reversal phenomenon, inter-tissue transcriptome divergence may be followed by increasing inter-tissue similarity, or convergence, during ageing.

## Inter-tissue divergence during development and convergence during ageing

We studied the inter-tissue divergence/convergence question using two approaches. In the first, we analysed how transcriptome-wide expression variation among tissues changes with age regardless of their age-related expression patterns in any particular tissue. To do this, for each individual, we calculated the coefficient of variation (CoV) across the four tissues for each commonly expressed gene (n = 15,063), which represents a measure of expression variation among tissues. Then, we assessed how such inter-tissue variation changes over the lifetime by calculating the Spearman's correlation coefficient between CoV and age separately for development and ageing periods (correlation values for all genes are given in *Figure 2—source data 1*).

Using the CoV values calculated across all 15,063 genes (excluding one 904-day-old individual for which we lacked the cortex data), we observed a significant mean CoV increase in development (Spearman's correlation coefficient $\rho$ = 0.77, two-sided p=0.041), confirming that tissues diverge as development progresses (*Figure 2a*). Interestingly, during ageing, we observed a decrease in mean CoV with age, albeit not significant ($\rho$ = −0.50, p=0.204, *Figure 2a*), suggesting that tissues may tend to converge during ageing. This was also supported by the PCA in which we observed a trend of ageing-associated decrease in mean Euclidean distance among tissues (using PC1–PC4 space with quantile-normalised data: $\rho$ = −0.87, p=0.0026; with VST-normalised data $\rho$ = −0.58, p=0.102, *Figure 1—source data 1*). We obtained the same divergence-convergence pattern by calculating the median CoV values for each individual instead of the mean (*Figure 2—figure supplement 1*). *Figure 2b* exemplifies this pattern of increasing and then decreasing CoV through lifetime for the gene displaying the strongest such signal.

We identified n = 9058 genes showing divergent trends among tissues in development based on their CoV change with age (without using a significance cutoff per gene). Among these, n = 4802 showed convergent trends in ageing, which we refer to as DiCo genes. We next studied the transition points between divergence and convergence by clustering genes showing the DiCo pattern (n = 4802) based on their CoV values (*Figure 2—figure supplement 2*). Notably, cluster 1, which shows a slightly delayed divergence starting after 8 days and peaks around 3 months, was associated with metabolic and respiration-related processes (FDR-corrected p-value<0.1), and cluster 5, which shows a relatively delayed convergence after 4 months, was enriched in categories related to vascular development (FDR-corrected p-value<0.1) (*Supplementary file 4*). To assess the contribution of different tissues to the DiCo pattern, we further clustered DiCo-displaying genes (n = 4802) based on their expression levels (*Figure 2—figure supplement 3*). Not surprisingly, the clusters with relatively higher expression levels of a tissue (e.g. muscle in cluster 9) were enriched in functional categories (FDR-corrected p-value<0.1) related to that tissue (e.g. muscle cell development) (*Supplementary file 5*).

We then studied DiCo at the single-gene level. We tested each gene for a significant CoV change in their expression levels (i.e. divergence or convergence) in development and ageing (Spearman's correlation test with FDR-corrected p-value<0.1). We found that the ratio of divergent and convergent genes differed significantly between development (70% divergence among 2581 significant genes) and ageing (68% convergence among 62 significant genes) (*Figure 2d and e*). The same pattern was also observed without using significance cutoff (*Figure 2—figure supplement 4*). We also confirmed that this pattern is also observed with VST-normalised data (Materials and methods), and is thus not affected by the data preprocessing approach (*Figure 2—figure supplement 14*).

To our knowledge, inter-tissue convergence during ageing is a novel phenomenon. We first considered the possibility that convergence during ageing could be explained by heteroscedasticity which could arise due to increased inter-individual variability in gene expression during ageing (*Somel et al., 2006*). To test this hypothesis, we compared expression-age heteroscedasticity levels between two

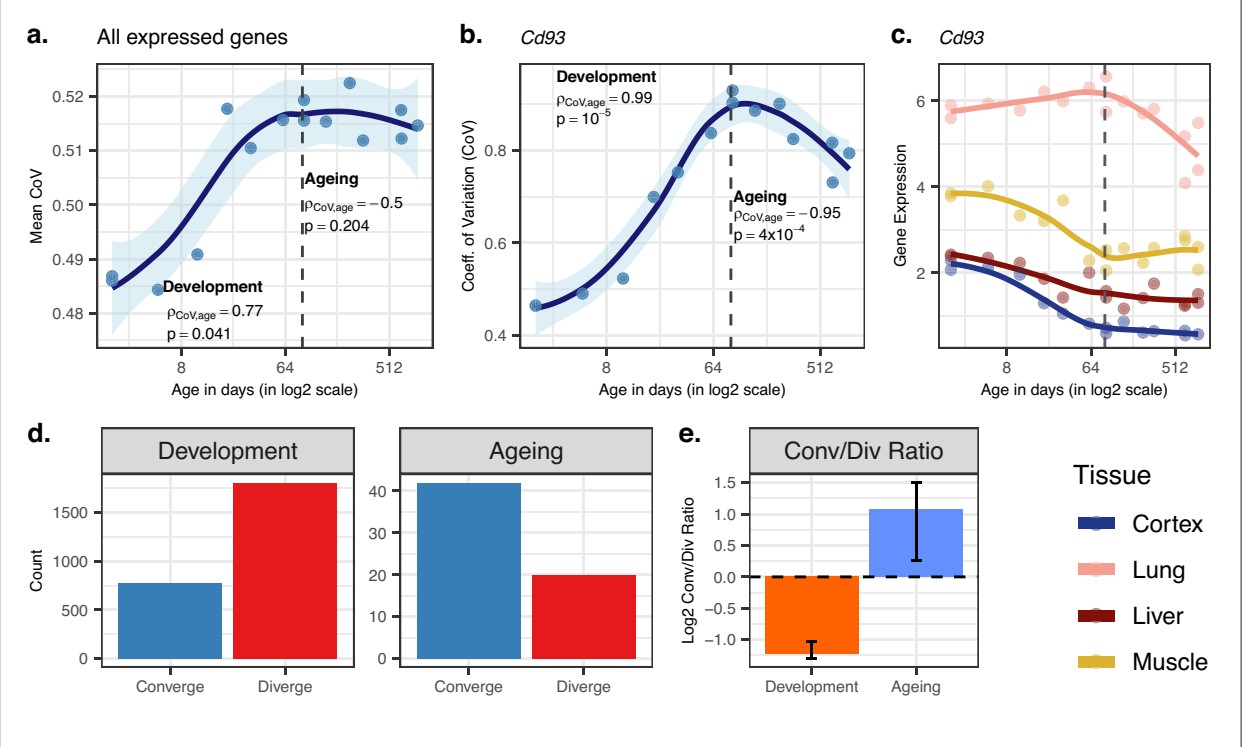

**Figure 2.** Age-related change in gene expression variation among tissues estimated with coefficient of variation (CoV). (**a**) Transcriptome-wide mean CoV trajectory with age. Each point represents the mean CoV value of all protein-coding genes (15,063) for each mouse (n = 15) except the one that lacks expression data in the cortex. (**b**) Age effect on CoV value of the Cd93 gene which has the highest rank for the divergence-convergence (DiCo) pattern in four tissues (Materials and methods). CoV increases during development and decreases during ageing, indicating expression levels show DiCo patterns among tissues. (**c**) Expression trajectories of the gene Cd93 in four tissues. (**d**) The number of significant CoV changes with age (false discovery rate [FDR]-corrected p-value<0.1) during development (left, $n_{converge}$ = 772, $n_{diverge}$ = 1809) and ageing (right, $n_{converge}$ = 42, $n_{diverge}$ = 20). Converge: genes showing a negative correlation ($\rho$) between CoV and age; diverge: genes showing a positive correlation between CoV and age. (**e**) Log2 ratio of convergent/divergent genes in development and in ageing. The graph represents only genes showing significant CoV changes (FDR-corrected p-value<0.1, given in panel **d**). Error bars represent the range of log2 ratios calculated from leave-one-out samples using the jackknife procedure (Materials and methods, values are given in *Figure 2—source data 1*).

The online version of this article includes the following source data and figure supplement(s) for figure 2:

**Source data 1.** All the data related to divergence-convergence (DiCo) pattern: age-related coefficient of variation (CoV) change of genes, pairwise tissue expression correlations, analysis of independent datasets; GSE34378 (Jonker et al.), GSE132040 (Schaum et al.), and GTEx.

**Figure supplement 1.** Age-related change in coefficient of variation (CoV) summarised across genes using median CoV values.

**Figure supplement 2.** Clustering of divergence-convergence (DiCo) genes by expression variations (coefficient of variation [CoV]) among tissues.

**Figure supplement 3.** Clustering of divergence-convergence (DiCo) genes by expression levels in tissues.

**Figure supplement 4.** Number of genes with inter-tissue divergence and convergence tendencies in development and ageing.

**Figure supplement 5.** Pairwise tissue expression correlations.

**Figure supplement 6.** Summary of pairwise expression correlations among tissues.

**Figure supplement 7.** Coefficient of variation (CoV) and pairwise correlation analysis of Jonker dataset.

**Figure supplement 8.** Principal components analysis (PCA) of GTEx dataset covering cortex, liver, lung, and muscle tissues.

**Figure supplement 9.** Coefficient of variation (CoV) and pairwise correlation analysis of GTEx dataset covering cortex, liver, lung, and muscle tissues.

**Figure supplement 10.** Principal components analysis (PCA) of GTEx dataset with 10 tissues.

**Figure supplement 11.** Coefficient of variation (CoV) and pairwise correlation analysis of GTEx dataset with 10 tissues.

**Figure supplement 12.** Permutation test result for the proportion of divergence-convergence (DiCo) genes.

**Figure supplement 13.** Clustering of tissues by the presence of samples from the same individuals.

**Figure supplement 14.** Reproducing *Figure 2* results with variance stabilising transformation (VST) normalisation.

**Figure supplement 15.** Effect of heteroscedasticity to divergence-convergence (DiCo) pattern.

*Figure 2 continued on next page*

*Figure 2 continued*

**Figure supplement 16.** Sex effect on coefficient of variation (CoV) analysis using GTEx.

**Figure supplement 17.** Principal components analysis (PCA) of Schaum dataset covering cortex, liver, lung, and muscle tissues.

**Figure supplement 18.** Coefficient of variation (CoV) and pairwise correlation analysis of Schaum dataset covering cortex, liver, lung, and muscle tissues.

**Figure supplement 19.** Principal components analysis (PCA) of Schaum dataset with eight tissues.

**Figure supplement 20.** Coefficient of variation (CoV) and pairwise correlation analysis of Schaum dataset with eight tissues.

gene sets: (1) genes with the DiCo pattern and (2) genes showing divergent patterns throughout lifetime (divergent-divergent [DiDi, n = 4182]) for each tissue separately (Materials and methods). We did not observe any significant difference in heteroscedasticity between DiCo and DiDi genes in any of the tissues (two-sided Kolmogorov–Smirnov [KS] test, p>0.05 in all tissues, *Figure 2—figure supplement 15*), which suggests that heteroscedasticity due to increased inter-individual variability probably does not drive the observed age-related convergence during ageing. Visual inspection of gene expression clusters also suggested that the DiCo pattern is not particularly associated with nonlinear changes in gene expression with age (*Figure 1—figure supplements 12–15*).

In order to further verify the DiCo pattern, we used a second approach to test it in our mouse dataset. For each individual, we calculated correlations between pairs of tissues across their gene expression profiles. Under the DiCo pattern, we would expect pairwise correlations to decrease during development and increase during ageing. Among all pairwise comparisons, we observed a strong negative correlation during development ($\rho$ = [-0.61, –0.9], nominal p<0.05 in five out of six tests), while during ageing, four out of six comparisons showed a moderate positive correlation ($\rho$ = [0.16, 0.69], nominal p<0.05 in one out of six comparisons, *Figure 2—figure supplement 5*). Calculating the mean of pairwise correlations among tissues for each individual, we observed the same DiCo pattern (nominal p<0.05 for both periods, *Figure 2—figure supplement 6*).

## The DiCo pattern indicates loss of tissue specificity during ageing

Potential explanations of the DiCo pattern involve two scenarios consistent with the age-related loss of identity: (1) decreased expression of tissue-specific genes in their native tissues or (2) non-specific expression of tissue-specific genes in other tissues. To test these predictions, we first identified tissue-specific gene sets based on relatively high expression of that gene in a particular tissue (cortex: 1175; lung: 839; liver: 986; muscle: 766 genes). We noted that tissue-specific genes show clear up-down reversal patterns, being mostly upregulated during development, and downregulated during ageing (*Figure 3*, 57–89%). The up-down reversal pattern was particularly strong among tissue-specific genes for the three of four tissues tested (OR = [1.65, 6.52], p<0.05 for each tissue except in liver: OR = 0.87, p=0.09, *Figure 3—source data 1*). Tissue-specific genes were also enriched among DiCo genes (*Figure 3—source data 1*, OR = 1.56, Fisher's exact test p<10$^{-16}$).

We then tested our initial prediction that the DiCo pattern is related to tissue-specific genes losing their expression in their native tissue and/or gaining expression in non-native tissues during ageing. We first tested this hypothesis by considering all tissue-specific genes. We found a positive odds ratio between loss of expression in native tissue and gain in other tissues during ageing (OR = 5.50, Fisher's exact test p=2.1 × 10$^{-129}$, *Figure 4a*). The same analysis conducted with only the DiCo genes yielded a much stronger association (OR = 74.81, Fisher's exact test p=5.9 × 10$^{-203}$, *Figure 4b*). This suggests that loss of tissue-specific expression is observed across the transcriptome, with a particularly strong association among DiCo genes. *Figure 4c–f* exemplifies the expression trajectories of genes chosen from each group defined in *Figure 4b*.

We then asked whether genes displaying the DiCo pattern may be related to specific functional pathways or share specific regulators. Using GO, we searched for functional enrichment among convergent genes during ageing using developmentally divergent genes as the background (Materials and methods). We found enrichment for 184 GO Biological Process (BP) categories for the DiCo pattern (KS test, FDR-corrected p-value<0.1, *Figure 4—source data 1*) and summarised enriched categories by clustering them based on the number of genes they share. We then studied the trends of gene expression changes with age (without a significance cutoff) in each representative category for each tissue (Materials and methods) (*Figure 4h*; we provide detailed clustering for the categories

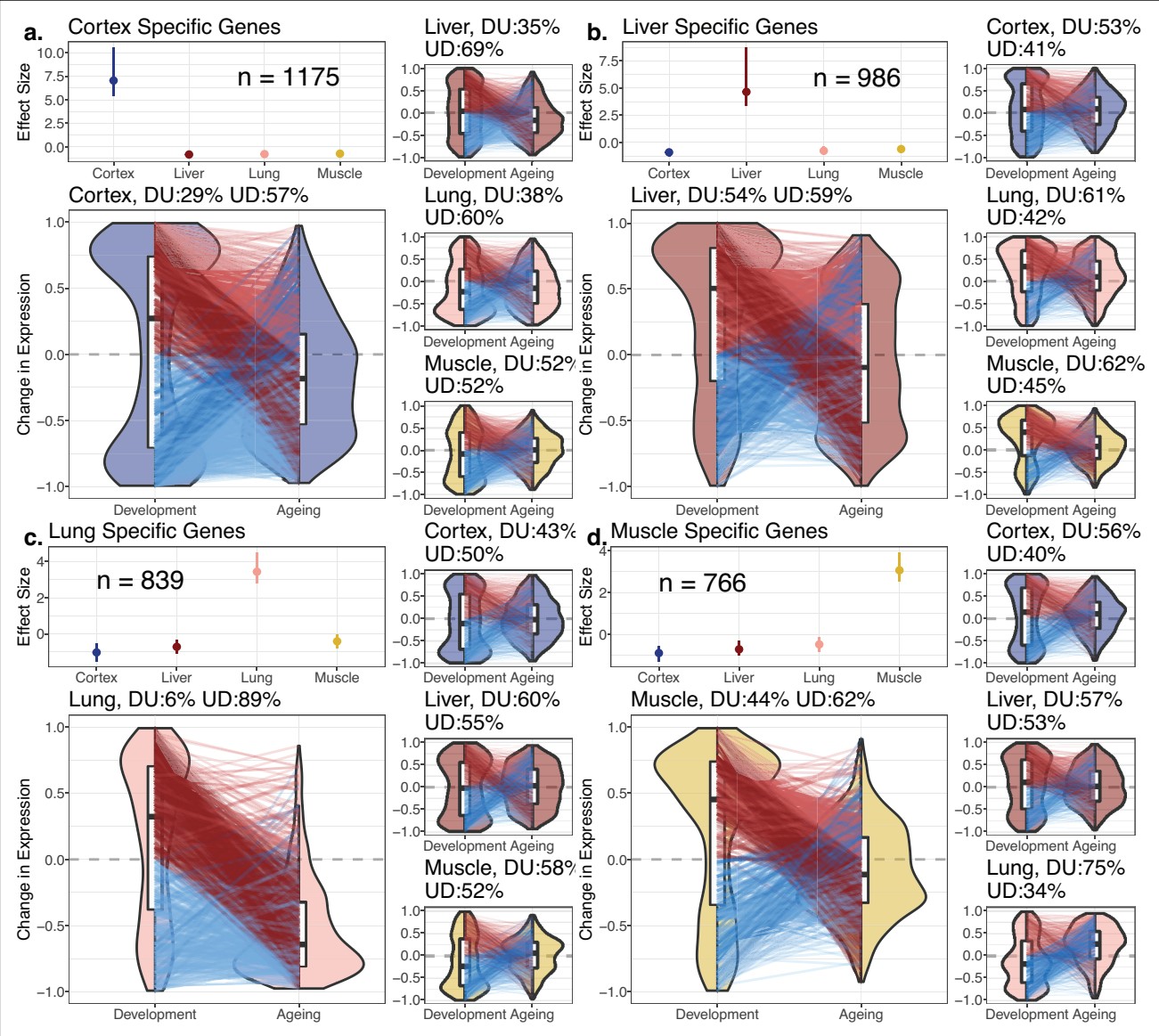

**Figure 3.** Reversal patterns among tissue-specific genes. Age-related expression changes of the tissue-specific genes. In each panel (**a–d**), the upper-left subpanels show effect size (ES) calculated with the Cohen's d formula, using expression levels of each gene among tissues (Materials and methods). The IQR (line range) and median (point) ES for each tissue are shown. The number of tissue-specific genes is indicated inside each subpanel. The lower-left subpanels show violin plots of the distribution of age-related expression change values (Materials and methods) among tissue-specific genes in development and in ageing. Each quadrant represents the plots for each tissue-specific gene group. The red and blue lines connect gene expression changes for the same genes in development and ageing. DU: percentage of down-up reversal genes among downregulated, tissue-specific genes in development; UD: percentage of up-down reversal genes among upregulated, tissue-specific genes in development. Tissue-specific genes are enriched among UD reversal genes except in the liver (Fisher's exact test; $OR_{cortex}$ = 1.65, $OR_{lung}$ = 6.52, $OR_{liver}$ = 0.87, $OR_{muscle}$ = 1.26, $p<0.05$ for each test except in liver).

The online version of this article includes the following source data for figure 3:

**Source data 1.** Effect sizes for determination of tissue-specific genes, enrichment of divergence-convergence (DiCo), and reversal genes within tissue-specific genes.

in 'Other GO' **Figure 4—figure supplement 1**). On average, energy metabolism, mitochondria, and tissue function-related categories, as well as immune response-related categories, exhibit DiCo-type expression changes over time and across tissues, where temporal changes in different tissues occur in opposite directions. Notably, for the majority of representative GO categories, the lung had the most distinct expression patterns in both periods (**Figure 4h**, **Figure 4—figure supplement 1**).

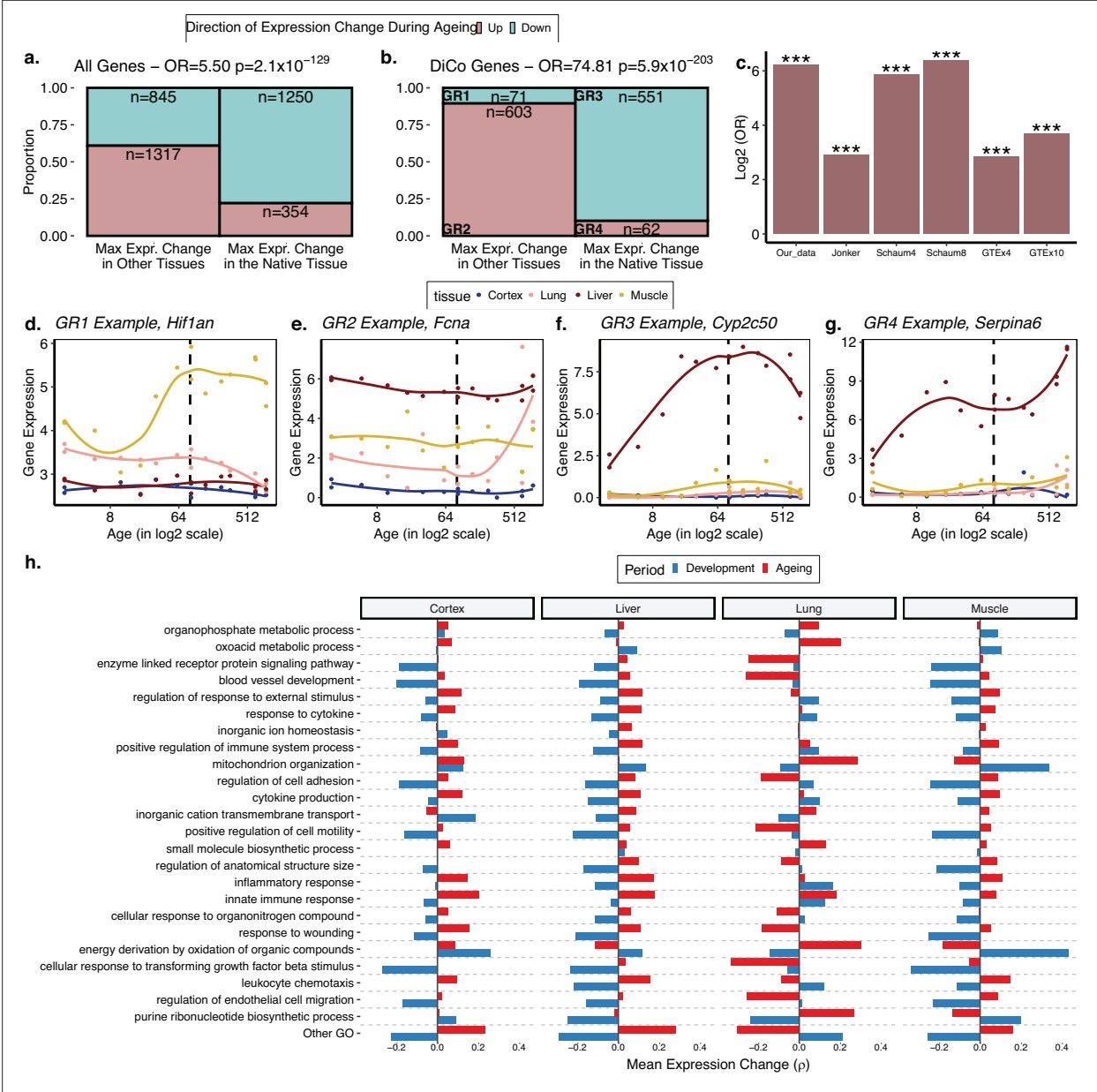

**Figure 4.** The loss of tissue-specific expression during ageing and functional enrichment of divergence-convergence (DiCo) genes. (**a**) Mosaic plot showing the association between maximal expression change in native vs. non-native tissues (x-axis) vs. down- (cyan) or upregulation (pink) during ageing across all tissue-specific genes (n = 3766). The highly significant odds ratio indicates that genes native to a tissue tend to be downregulated during ageing in that native tissue if they show maximal expression change during ageing in that tissue. Conversely, if they show maximal expression change during ageing in non-native tissue, those genes are upregulated during ageing. Consequently, tissue-specific expression patterns established during development will tend to be lost during ageing. (**b**) The same as (**a**) but using only the tissue-specific genes that show the DiCo pattern (n = 1287). (**c**) Summary of the association tests for 'direction of maximal expression change in native vs. non-native tissues' across all datasets analysed. The y-axis shows log2-transformed odds ratio (OR) for each dataset (x-axis) – Schaum4: using the same four tissues as our dataset. Schaum8: using eight tissues. GTEx4: using the same four tissues as our dataset. GTEx10: using 10 tissues. ***False discovery rate (FDR)-corrected p-value<10$^{-87}$. p-Values are given in **Table 2**. The four groups are annotated as GR1–4 and gene expression changes for each group in our dataset is exemplified in (**d–g**). (**h**) Trends of expression change with age of genes (x-axis) in categories enriched in DiCo (Gene Set Enrichment Analysis [GSEA]). Enriched categories (n = 184) are summarised into representatives (y-axis) using hierarchical clustering and Jaccard similarities (Materials and methods). Categories are ordered by the number of genes they contain from highest (bottom, n = 290) to lowest (top, n = 26). The most distant cluster with low within-cluster similarity in the hierarchical clustering (other Gene Ontology [GO]) was clustered separately and given in **Figure 4—figure supplement 1**.

The online version of this article includes the following source data and figure supplement(s) for figure 4:

**Source data 1.** Gene Set Enrichment Analysis (GSEA) result of divergence-convergence (DiCo) genes, DiCo enrichment with tissue-specific expression

*Figure 4 continued on next page*

*Figure 4 continued*

loss, age-related expression change correlations, and convergence overlaps among datasets.

**Figure supplement 1.** Age-related expression change trends in divergence-convergence (DiCo)-enriched categories denoted as 'Other GO' in the first clustering.

**Figure supplement 2.** Comparison of datasets.

Contrary to the functional enrichment results, we did not find any specific regulators (miRNA or transcription factors) associated with DiCo using the same background as above (at 235 tests for miRNA and 158 tests for TF, FDR-corrected p-value>0.1 for both tests) (Materials and methods), which suggests that DiCo pattern may not be driven by a limited number of specific regulators, but may instead be a transcriptome-wide phenomenon.

## Additional mouse and human datasets confirm the association between loss of tissue specificity and inter-tissue convergence during ageing

We investigated inter-tissue convergence during ageing in three additional datasets where multiple tissue samples were available for the same individuals (*Table 1*). We conducted the analysis using a subset of the same four tissues in our dataset and also larger sets when additional samples were available. Age-related expression changes showed small to moderate correlations among all datasets analysed, with our dataset being most similar to the mouse dataset from Jonker et al., while the GTEx human dataset was the most distinct (*Figure 4—figure supplement 2a*).

First, using the Jonker et al. dataset (*Jonker et al., 2013*) comprising five tissues (*Table 1*), we observed transcriptome-wide convergence during ageing with a significant decline in mean Euclidean distance between PCs ($\rho$ = –0.57, p=0.014, *Figure 2—figure supplement 7a–c*) and a strong decrease in mean CoV during ageing ($\rho$ = –0.48, p=0.044, *Figure 2—figure supplement 7d*). Moreover, we found that 7/10 tissue pairs showed increased pairwise tissue correlations during ageing, although none of them was significant after multiple testing correction (*Figure 2—figure supplement 7f*). 66% of the genes with a significant change in CoV were convergent comparable to our dataset showing 68% convergence among significant changes. We also tested the association between the loss of identity and convergence pattern by repeating the same analysis as in *Figure 4b* with the

**Table 1.** Dataset characteristics summarising species, tissues, number of individuals, age range, sex, and platform used for measuring gene expression values.

| Dataset | Species | Tissues | N | Age range | Sex | Method |
|---|---|---|---|---|---|---|
| Izgi et al., four tissues | Mice | Brain, lung, liver, muscle | 8 | 3–30 months | Male | RNA-seq |
| Jonker et al., five tissues | Mice | Brain, lung, liver, kidney, spleen | 18 | 3–30 months | Female | Microarray |
| Schaum et al., four tissues | Mice | Brain, lung, liver, muscle | 37 | 3–27 months | Male (n = 26) Female (n = 11) | RNA-seq |
| Schaum et al., eight tissues | Mice | Brain, lung, liver, muscle, subcutaneous fat, kidney, heart, spleen | 26 | 3–27 months | Male (n = 20) Female (n = 6) | RNA-seq |
| GTEx, four tissues | Humans | Brain, lung, liver, muscle | 47 | 20–75 years | Male (n = 36) Female (n = 11) | RNA-seq |
| GTEx, 10 tissues | Humans | Adipose, tibial artery, cerebellum, lung, skeletal muscle, tibial nerve, pituitary, sun-exposed skin, thyroid, whole blood | 35 | 20–75 years | Male (n = 27) Female (n = 8) | RNA-seq |

**Table 2.** Result summary of the all datasets analysed.

First column shows the names of datasets analysed. Numbers in parentheses show the sample sizes. 'Among all genes' column refers to the analyses performed using all genes relevant to those analyses (subcolumns) without a significance cutoff. 'Within significant CoV changes': genes show significant CoV change with age with FDR-corrected p-value<0.1. In the 'DiCo vs. tissue specificity (Di- as background)' column, divergent genes in development (Di-) were chosen as background. 'Co vs. expression change in native tissue association (*Figure 4b*)' column refers to the analysis performed in *Figure 4b* for each dataset, and the results are presented in *Figure 4c*. The association tests were performed among convergent genes in ageing except in our dataset, which was performed with DiCo genes. Significant test results are indicated with italic fonts. Bold fonts show the results that support convergence or tissue-specific expression loss in ageing whether as a significant result or as a trend. Unsupportive test results and inapplicable tests are written in normal font.

| | Among all genes | | | | | Within significant CoV changes |
|---|---|---|---|---|---|---|
| | PCA change in Euclidean distance | Mean CoV change | Median CoV change | Pairwise tissue correlations | DiCo vs. tissue specificity (Di- as background) | Co vs. expression change in native tissue association (Figure 4b) | Co vs. Di proportions |
| **Izgi2022** | $\rho = -0.87$, p=0.0026 | $\rho = -0.5$, p=0.2 | rho = $-0.48$, P = 0.23 | 4/6 positive, none significant* | OR = 1.56, p=1.3 × 10$^{-18}$ | *OR = 74.81, p=5.9 × 10$^{-203}$ (among 1287 DiCo genes)* | 68% convergence (among 62 significant genes*) |
| Jonker 2013 five tissues, two different than ours (n = 18) | $\rho=-0.57$, p=0.014 | $\rho=-0.48$, p=0.044 | $\rho = -0.03$, p=0.91 | 7/10 positive, none significant* | Di- background missing | *OR = 7.52, p=6.5 × 10–109 (among 2967 convergent genes)* | 66% convergence (among 1735 significant genes*) |
| Schaum 2020, same four tissues (n = 37) | $\rho = 0.13$, p=0.46 | $\rho = 0.25$, p=0.14 | $\rho = 0.13$, p=0.43 | 4/6 positive, two significant* | *OR = 1.33, p=1.07 × 10$^{-8}$* | *OR = 58.03, p=1.5 × 10$^{-197}$ (among 2124 convergent genes)* | 53% convergence (among 319 significant genes*) |
| Schaum 2020, eight tissues (n = 26) | $\rho = 0.1$, p=0.62 | $\rho = 0.16$, p=0.43 | $\rho = 0.04$, p=0.86 | 16/28 positive, five significant* | Di- background missing | *OR = 84.2, p=9.7 × 10$^{-96}$ (among 2380 convergent genes)* | 54% convergence (among 244 significant genes*) |
| GTEx, same four tissues | $\rho = -0.23$, p=0.12 | $\rho = -0.12$, p=0.42 | $\rho = -0.18$, p=0.23 | 5/6 positive, none significant* | Di- background missing | *OR = 7.21, p=7 × 10$^{-87}$ (among 2407 convergent genes)* | (no significant CoV changes) |
| GTEx, 10 tissues | $\rho = -0.26$, p=0.13 | $\rho = -0.14$, p=0.44 | $\rho = -0.3$, p=0.08 | 29/45 positive, none significant* | Di- background missing | *OR = 13.01, p=5.7 × 10$^{-114}$ (among 2195 convergent genes)* | (all three significant genes were convergent) |

$\rho$ = Spearman's correlation coefficient; OR = odds ratio; FDR = false discovery rate; CoV = coefficient of variation; DiCo = divergence-convergence; PCA = principal components analysis.

* FDR-corrected p-value<0.1.

Jonker et al. dataset using only the convergent genes in ageing as we lack developmental period. We again found strong association, consistent with convergent genes losing expression in their native tissue and gaining in other tissues during ageing (OR = 7.52, p<10$^{-16}$, *Figure 4c*). The results are summarised in *Table 2*.

Next, we used another mouse dataset by *Schaum et al., 2020* (*Table 1*). Repeating the analysis on the same four tissues and also a larger set of eight tissues, we did not find support for transcriptome-wide convergence (*Table 2*, *Figure 2—figure supplements 17 and 19*). In the 4-tissue comparison, 4/6 tissue pairs, and in the 8-tissue comparison only 16/28 tissue pairs showed positive correlations, supporting the inter-tissue convergence during ageing (*Figure 2—figure supplements 18c and 20c*). Interestingly, 75% of the negative correlations involved muscle and subcutaneous fat. Convergence ratios among genes showing significant change in CoV (FDR-corrected p-value<0.1) were marginally above 50%. Although we did not observe widespread convergence during ageing in this dataset, we still detected strong associations between convergence in ageing and tissue specificity (OR$_{4-tissue}$ = 1.33, p=1.08 × 10$^{-8}$) and identity loss (OR$_{4-tissue}$ = 58.3, p<10$^{-16}$; OR$_{8-tissue}$ = 84.2, p < 10$^{-16}$) (*Figure 4c*).

Lastly, we used the GTEx dataset to investigate inter-tissue convergence during ageing in humans. Calculating the change in mean Euclidean distance based on PCA and mean CoV values, we found a non-significant tendency towards convergence across the whole transcriptome in the same 4 tissues and a larger set of 10 tissues (*Table 2*, *Figure 2—figure supplements 8 and 10*). We also performed

the four-tissue comparison with female and male individuals separately and observed relatively strong inter-tissue convergence among ageing females ($\rho_{female}$ = –0.58, $p_{female}$ = 0.059) but less in males ($\rho_{male}$ = –0.052, $p_{male}$ = 0.77) which lack individuals at the youngest and oldest age groups (*Figure 2—figure supplement 16*). Moreover, 5/6 and 29/45 tissue pairs showed increased correlation with age in 4-tissue and 10-tissue comparisons, consistent with inter-tissue convergence during ageing (*Figure 2—figure supplements 9 and 11*). Notably, 8 of 16 negative correlations in the 10-tissue comparison involved the skin tissue (*Figure 2—figure supplement 11c*). We also studied significant changes in CoV per gene, but found no significant gene in the 4-tissue comparison and only three genes in the 10-tissue comparison, all of which were convergent. Finally, we tested the association between the loss of expression in native tissue and gain in other tissues during ageing among convergent genes, confirming the association with the tissue identity (*Figure 4c*, *Table 2*).

Overall, analysis of these three additional datasets indicates that inter-tissue convergence during ageing is commonly, but not always, observed at the transcriptome-wide level in mice and in humans. Notably, the transcriptome-wide trend was weak in the Jonker et al. and GTEx datasets and not evident in the Schaum et al. dataset. The association between the loss of identity and convergence, on the other hand, was strong across all datasets (*Table 2*).

We further asked whether convergent gene sets identified in different datasets overlap. 11 of 15 comparisons were significant, but the effect sizes (ESs) were small (*Figure 4—figure supplement 2b*). We reason that the low overlap across datasets might reflect that transcriptome-wide convergence was weak and that we lack the developmental samples for the external datasets, that is, we can only compare convergence during ageing but not the DiCo pattern. Noteworthy, only 62% of convergent genes in ageing are divergent during development in our dataset, and low overlap between convergence does not rule out overlap across DiCo genes.

These results suggest that inter-tissue convergence in ageing may be a weak but widespread phenomenon and associated with the loss of tissue identity. Overall, while mouse and human tissues display divergence in development (*Figures 1a and 2a*, *Cardoso-Moreira et al., 2019*), this appears to be followed by a trend towards inter-tissue convergence in ageing (*Figure 2a*, *Figure 2—figure supplements 1–20*) and could be linked to loss of tissue identity.

## Changes in cellular composition and cell-autonomous expression can both explain the DiCo pattern

Ageing-related transcriptome changes observed using bulk tissue samples may be explained by temporal changes in cell-type proportions within tissues, by cell-autonomous expression changes, or both. To explore whether the observed inter-tissue DiCo patterns may be attributed to changes in cell-type proportions, we used published data from a mouse single-cell RNA-sequencing experiment (*Tabula Muris Consortium, 2020*). For each of the four tissues in our original experiment, we collected cell-type-specific expression profiles from 3-month-old young adult mice in the Tabula Muris Senis dataset. We deconvoluted bulk tissue expression profiles in our mouse dataset using the corresponding tissue's cell-type-specific expression profiles by regression analysis (Materials and methods) and studied the relative contributions of each cell type to tissue transcriptomes and how these change with age. The analysis was performed with three gene sets; all genes (n = [12,492, 12,849]), DiCo (n = [4007, 4106]), and non-DiCo genes (n = [8485, 8743]). Studying these deconvolution patterns, we observed a weak but consistent trend involving the most common cell types in different tissues. For instance, analysing DiCo genes in the liver and lung, we found that the most common cell type's contribution (hepatocyte in the liver, and bronchial smooth muscle cell in the lung) tends to increase during development (Spearman's correlation coefficient $\rho_{liver}$ = 0.95, $\rho_{lung}$ = 0.81, nominal p<0.05). This contribution then decreases during ageing ($\rho_{liver}$ = –0.77, $\rho_{lung}$ = –0.86, nominal p<0.05) (*Figure 5a*, *Figure 5—figure supplement 1*). This pattern was also observed in muscle and cortex, albeit not significantly (*Figure 5a*, *Figure 5—figure supplement 1*). These changes most likely reflect shifts in cellular composition, some of which were demonstrated directly in mice using in situ RNA staining (*Schaum et al., 2020*). Repeating the analysis with non-DiCo genes resulted in highly similar patterns considering the most common cell types in tissues, except in muscle ageing in which the age-related decrease was significantly higher with DiCo genes than the non-DiCo genes (permutation test with resampling all genes, $p_{skeletal-muscle-satellite-cell}$ = 0.04) (*Figure 5a*, *Figure 5—figure supplement 1*, *Figure 5—figure supplements 2–5*). These results indicate that the observed cellular

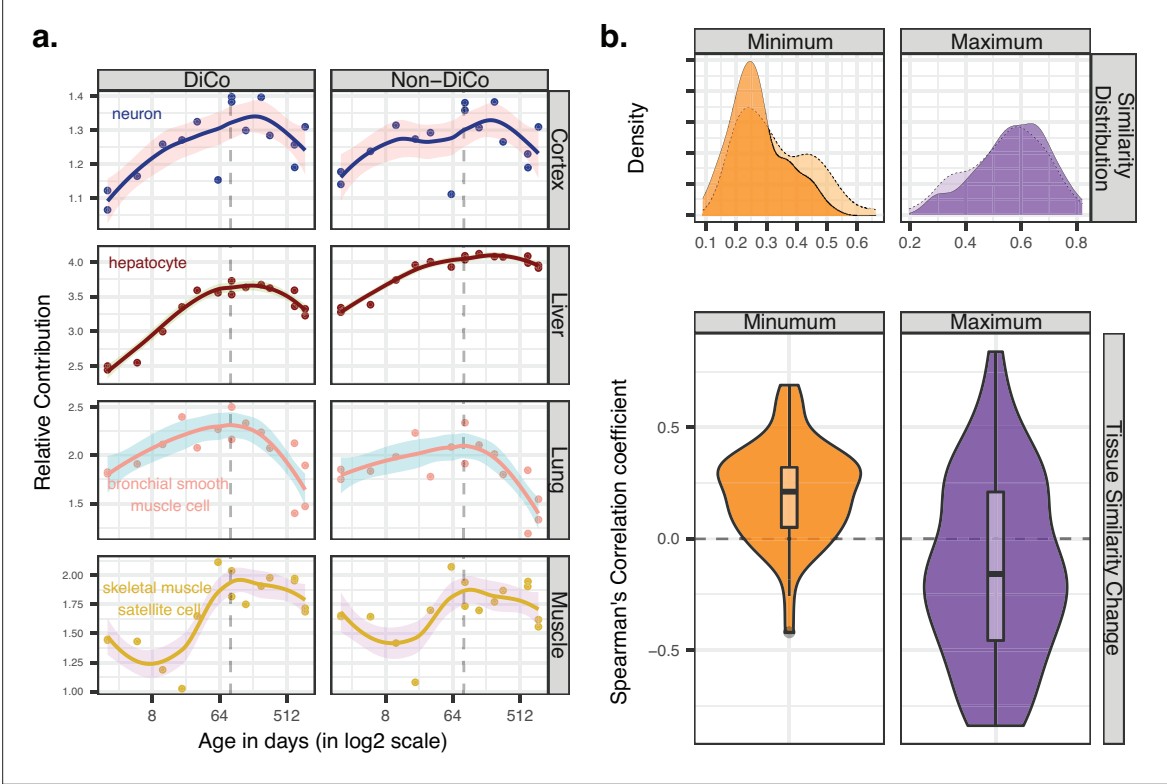

**Figure 5.** Contribution of tissue composition and cell-autonomous changes to the divergence-convergence (DiCo) pattern. (**a**) Deconvolution analysis of our mouse dataset with the 3-month-old scRNA-seq data (Tabula Muris Senis) using DiCo (n = [4007, 4106]) and non-DiCo (n = [8485, 8743]) genes. Only the cell types with the highest relative contributions to each tissue bulk transcriptome are shown (cell-type names are given within each plot). Contributions of all cell types to bulk tissue transcriptomes are shown in *Figure 5—figure supplement 1*. (**b**) Distribution of correlations for minimally (left) and maximally (right) correlated cell-type pairs among tissues (n = 54 pairs). For each cell type of a given tissue, one minimally (or maximally) correlated cell type is chosen from other tissues among the 3-month age group of the Tabula Muris Senis dataset (density plots with solid line edges). Dashed lines show the correlation distributions in 24-month age of minimally or maximally correlated cell-type pairs identified in the 3-month age group. Bottom panel shows age-related expression similarity ( $\rho$ ) changes of minimally (left) and maximally (right) correlated cell-type pairs. The correlation between age and tissue similarity (expression correlations) was calculated for each pair of cell types identified in the 3-month age group. All pairwise cell-type correlations and their age-related changes are given in *Figure 5—source data 1*.

The online version of this article includes the following source data and figure supplement(s) for figure 5:

**Source data 1.** Cell-type proportion estimation and cell-autonomous changes using the Tabula Muris Senis dataset.

**Figure supplement 1.** Age-related changes in cell-type proportions calculated using divergence-convergence (DiCo) and non-DiCo genes.

**Figure supplement 2.** Permutation-based comparison between divergence-convergence (DiCo) and non-DiCo-related cell-type proportion changes with age in the cortex.

**Figure supplement 3.** Permutation-based comparison between divergence-convergence (DiCo) and non-DiCo-related cell-type proportion changes with age in the liver.

**Figure supplement 4.** Permutation-based comparison between divergence-convergence (DiCo) and non-DiCo-related cell-type proportion changes with age in the lung.

**Figure supplement 5.** Permutation-based comparison between divergence-convergence (DiCo) and non-DiCo-related cell-type proportion changes with age in the muscle.

**Figure supplement 6.** Intra-tissue coefficient of variation (CoV) changes between cell types using the Tabula Muris Senis dataset.

composition changes may partly explain DiCo, although the influence of composition changes is not exclusive to genes displaying the DiCo pattern.

Next, we investigated the possible role of cell-autonomous changes in the DiCo pattern. Cell-autonomous changes could contribute to inter-tissue convergence during ageing in two ways. First, expression profiles of similar cell types shared across different tissues, such as immune cells, might converge with age. Another possible scenario, consistent with the notion of age-related cellular

identity loss, is that the expression profiles of unrelated cell types, such as tissue-specific cell types in different tissues, converge with age. To test these scenarios, we first ordered the pairwise correlations between cell types in different tissues at the 3-month age group to determine the most similar and dissimilar cell types across tissues (Materials and methods). Then, we studied how these similarities (i.e. pairwise correlations) change with age (*Figure 5b*). Intriguingly, we found that pairs of similar cell types (i.e. those with the highest correlations) among tissues tend to become less similar with age (36/54 [67%] of pairwise comparisons, *Figure 5—source data 1*). On the contrary, the most distinct cell types (i.e. those with the lowest correlations) among tissues become more similar with age (45/54 [83%], *Figure 5—source data 1*). Repeating the analysis considering DiCo genes only yielded a similar trend (30/54 [56%] decrease in correlation among the most similar cell types, permutation test with resampling non-DiCo genes, p>0.1; and 47/54 [87%] increase in correlation among the most distinct cell types, permutation test, p>0.1). These trends are consistent with age-related cellular identity loss, and they suggest that cell-autonomous changes may also contribute to inter-tissue convergence during ageing, although further data and analyses would be needed to fully establish their validity.

Finally, we tested the possibility of intra-tissue convergence of cell types in the Tabula Muris Senis dataset by calculating expression variation among cell types using the CoV measure for each individual. However, we did not observe a consistent trend of increasing similarity among cell types within tissues from 3-month-old to 24-month-old mice (*Figure 5—figure supplement 6*).

## Discussion

Our findings confirm a number of ageing-associated phenomena identified earlier, while also revealing new patterns. First, we report parallel age-related expression changes among the four tissues studied, during development, as well as in ageing. The inter-tissue correlation distributions were modest and also comparable between development and ageing (*Figure 1c*). This last point may appear surprising at first glance, given the stochastic nature of ageing relative to development (*Bahar et al., 2006*; *Martinez-Jimenez et al., 2017*; *Angelidis et al., 2019*; *Somel et al., 2006*; *Feser et al., 2010*; *Kim et al., 1996*; *Enge et al., 2017*), and also given earlier observations that developmental expression changes tend to be evolutionarily conserved, while ageing-related changes much less so (*Zahn et al., 2007*; *Somel et al., 2010*). At the same time, when we consider that tissues diverge during development, and also that ageing is characterised by parallel expression changes among tissues related to damage response, inflammation, and reduced energy metabolism (*Zahn et al., 2007*; *Yang et al., 2015*), similar magnitudes of correlations during development and ageing may be expected.

Second, we verify the generality of the reversal pattern, that is, up-down or down-up expression change patterns across the lifetime, among distinct mouse tissues that include both highly mitotic (lung and liver) and less mitotic ones (skeletal muscle and cortex). Consistent with earlier observations in fewer tissues (*Anisimova et al., 2020*; *Dönertaş et al., 2017*), we find that about half the expressed genes display reversal in all cases studied. Importantly, expression reversal is not ubiquitous across all genes and our findings do not necessarily contradict the hyperfunction theory. Instead, we suggest that reversal is a common phenomenon that influences a notable fraction of the transcriptome and is a likely contributor to mammalian ageing.

Two observations here are notable. One is that reversal-displaying genes, especially those displaying the up-down pattern in each tissue, can be associated with tissue-specialisation-related pathways (e.g. morphogenesis) and tissue-specific functions (e.g. synaptic activity). The second observation is the lack of significant overlap among reversal genes among tissues. We thus hypothesised that reversals might be reflecting tissue specialisation during development (hence lack of overlap among tissues) and loss of specialisation during ageing. These processes could manifest themselves as inter-tissue divergence and convergence patterns over lifetime. We indeed observed that the up-down reversal pattern is enriched in tissue-specific genes, except in the liver. Studying inter-tissue similarity across mouse lifespan, we further found that the four tissues' transcriptomes diverged during postnatal development, and we further detected a trend towards inter-tissue convergence during ageing. We then further investigated this phenomenon through different approaches: (1) by studying overall trends using PCA, (2) by analysing transcriptome-wide trends of inter-tissue CoV without considering gene-wise significance cutoffs, (3) by focusing on genes with significant age-related changes in inter-tissue CoV, (4) by studying age-related changes in pairwise tissue correlations, (5) by analysing different cell types using scRNA-seq data, and (6) by repeating

the same analysis using independent mouse and human ageing datasets. The patterns we found were mostly consistent with inter-tissue convergence, but the majority of transcriptome-wide results were associated with low ESs, and some were not statistically significant. Importantly, all significant results suggested convergence during ageing. We therefore conclude that (1) developmental inter-tissue divergence does not continue into ageing and (2) convergence during ageing may be common although possibly not ubiquitous.

The weakness of the inter-tissue convergence signal per dataset and the limited overlap between convergent gene sets among datasets could have multiple reasons. These include the low signal-to-noise ratios characterising ageing-related expression patterns, the lack of old-age individuals in our mouse dataset (>3-year-old mice) and the GTEx dataset (>90-year-old humans), limited overlap of tissues between our mouse dataset (cortex, liver, lung, and muscle) and the Jonker et al. dataset (cortex, liver, lung, spleen, kidney), as well as differences in ageing patterns between species or between sexes. Further research involving larger sample sizes and diverse species is needed to confirm the generalisability of the observations.

Finally, we report a number of interesting observations on DiCo. We determine that tissue-specific genes tend to be downregulated in the tissues that they belong to during ageing, while non-tissue-specific genes are upregulated, which was confirmed by all external datasets (*Figure 4c*). Second, using deconvolution, we infer that cell types most common in a tissue (e.g. hepatocytes in the liver) tend to increase in frequency during development, but then decrease in frequency during ageing, as also shown recently using immunohistochemistry in a number of mouse tissues (*Schaum et al., 2020*). Accordingly, the DiCo phenomenon may at least partly be explained by shifts in cellular composition. This is intriguing as both highly mitotic and low mitotic tissues share this trend, indicating that an explanation based on stem cell exhaustion may not be applicable here. Third, we find increased expression similarity between distinct cell types in different tissues during ageing, but decreased similarity between similar cell types. Cell-autonomous expression changes, therefore, likely also contribute to the DiCo phenomenon. We note that higher expression variability among cells at old age (*Hernando-Herraez et al., 2019*; *Enge et al., 2017*) could also lead to inter-tissue convergence during ageing. A fourth interesting observation was the absence of significant enrichment for specific transcription factor or microRNA targets among DiCo genes. This result may not be surprising if inter-tissue convergence is mostly driven by stochastic damage accumulation, such as loss of epigenetic marks. It is also possible that instead of specific regulators their interaction and cooperativity are associated with the DiCo. Future experimental studies could test both mechanistic aspects and functional link to tissue specificity.

We also note two major limitations of our study. One is related to the fact that our dataset represents bulk tissue samples, which may suffer from infiltration of foreign cell types into tissues. Indeed, one of the external datasets, Schaum et al., included samples from perfused mice (*Schaum et al., 2020*), and we did not find support for the transcriptome-wide convergence during ageing, even though the association between tissue identity loss and convergence was also evident. The scRNA-seq dataset we analysed further suggested that DiCo is associated with tissue-specific genes and not immune- or blood-related categories, but we still cannot rule out possible infiltration artefacts that may affect our results. A second limitation is related to ageing being highly sex-dimorphic in mammals (*Yuan et al., 2012*; *Sampathkumar et al., 2020*). Hence, in-depth analysis of sex specificity of the DiCo pattern could be relevant. Our mouse dataset included only male mice, while that of Jonker et al. was female-only. The fact that both revealed DiCo patterns suggest DiCo is not particular to one sex, but there could still exist sex-specific effects. In fact, when we analysed DiCo among human male and female individuals in the GTEx dataset separately, we observed slightly stronger inter-tissue convergence among ageing females than in males, although the GTEx male samples have also a drastically narrower age range (*Figure 2—figure supplement 16*). Accordingly, the prevalence of DiCo among humans and sexes waits to be determined.

Despite the open questions that remain, our results consistently support a model where ageing mammals suffer from loss of specialisation at the tissue level, and possibly also at the cellular level, which are observed as expression reversals and the newly discovered DiCo phenomenon we report here.

# Materials and methods

## Sample collection

We collected bulk tissue samples from 16 male C57BL/6J mice. The samples were snap frozen in liquid nitrogen and stored at –80°C. No perfusion was applied. The mice were of different ages covering the whole lifespan of *Mus musculus,* comprising both postnatal development and ageing periods. The samples included four different tissues; cerebral cortex, liver, lung, and skeletal muscle. One 904-day-old mouse had no cortex tissue sample and was thus excluded from the analysis. As a result, we generated 63 RNA-seq libraries in total.

## Separation of development and ageing periods

In order to compare gene expression changes during postnatal development and ageing, we studied the samples before sexual maturation (covering 2–61 days of age, n = 7) as the postnatal development period, and samples covering 93–904 days (n = 9 in all tissues except in cortex where we had n = 8) as the ageing period.

## RNA-seq library preparation

RNA sequencing was performed as previously described (*Liu et al., 2016*) with slight modifications. Briefly, total RNA was extracted using the Trizol reagent (Invitrogen) from frozen tissue samples. For sequencing library construction, we randomised all samples to avoid batch effects and used the TruSeq RNA Sample Preparation Kit (Illumina) according to the manufacturer's instruction. Libraries were then sequenced on the Illumina HiSeq 4000 system in three lanes within one flow cell using the 150 bp paired-end module.

## RNA-seq data preprocessing

The quality assessment of the raw RNA-seq data was performed using FastQC v.0.11.5 (*Andrews, 2010*). Adapters were removed using Trimmomatic v.0.36 (*Bolger et al., 2014*). The low-quality reads were filtered using the parameters: 'PE ILLUMINACLIP: TruSeq3-PE-2.fa:2:30:1:0:8:true, SLIDING-WINDOW:4:15, MINLEN:25'. The remaining high-quality reads were aligned to the mouse reference genome GRCm38 using STAR-2.5.3 (*Dobin et al., 2013*) with parameters: '`--sjdbOverhang 99 --outSAMattrIHstart 0 --outSAMstrandfield intronMotif --sjdbGTFfile GRCm38.gtf`'. The percentage of uniquely mapped reads in libraries ranged from 80% to 93%. We used cufflinks v.2.2.1 (*Trapnell et al., 2010*) to generate read counts for uniquely aligned reads (samtools '-q 255' filter) and calculated expression levels as fragment per kilobase million (FPKM). In total, we quantified expression levels for 51,608 genes in the GRCm38.gtf file. We identified 50 duplicated genes with 1 > FPKM value assigned, and the sum of their FPKM values was used.

All the remaining analyses were performed in R v.4.1. We restricted the whole analysis to only protein-coding genes obtained by the 'biotype' feature of the biomaRt library v.2.48.2 (*Durinck et al., 2009*). We also excluded genes which were not detected (0 FPKM) in 25% or more of the samples (at least 15 of 63), resulting in 15,063 protein-coding genes in total. As FPKM normalisation does not effectively account for cross-library variability, we additionally performed two normalisation approaches:

1. Quantile normalisation: Using all the samples together (n = 63, regardless of their age or tissue), FPKM values were log2 transformed (after adding 1) and quantile normalised with 'normalize.quantiles' function from 'preprocessCore' library v.1.54 (*Bolstad, 2020*). This approach equalises the distributions of different libraries. The assumption is that any large-scale differences in expression-level distributions reflect technical factors.

2. VST: To assess the robustness of quantile normalisation on downstream analysis, we additionally implemented this approach, which ensures homoscedasticity, that is, variances of expression levels are independent of the mean (*Anders and Huber, 2010*). Uniquely aligned reads obtained from the STAR alignment were used to calculate read counts by HTSeq v.0.13.5 (*Anders et al., 2015*) with parameters: '--format= bam --order= pos --stranded= no --type= exon --mode= union --nonunique= none'. Read counts were then imported into R using the 'DESeqDataSet-FromHTSeqCount' function in DESeq2 v.1.32.0 package (*Love et al., 2014*). The same filtration steps were applied as above, resulting in 14,973 protein-coding genes in total. Normalisation was performed with the 'vst' function and 'blinded = T' option in the DESeq2 package. The

VST-normalised expression matrix was used to reproduce results of *Figures 1 and 2*, which are given in *Figure 1—figure supplements 10 and 11* and *Figure 2—figure supplement 14*.

## Principal components analysis

We studied the main sources of variation in the whole dataset using PCA on the scaled expression matrix with 'prcomp' function in the R base. The first four components, PC1–PC4, explained 31, 20, 17, and 8% of the total variance. We observed a clear separation of tissues in PC1 and PC2 and a strong age effect in PC4. To statistically confirm tissue differences, we performed ANOVA on individual PC scores with tissue as explanatory variable; this was run on each of the first four PCs (PC1–PC4) separately. The magnitude of the age effect on PCA was measured with Spearman's correlation test between individual age and each individual's PC score separately in each tissue. PCA was also repeated for development and ageing periods separately (*Figure 1—figure supplement 3*). We further calculated Euclidean distance in pairwise manner among tissues of each individual in PC1–4 space constructed in three different ways: (1) using all the samples together, (2) using only the developmental samples, and (3) using only the ageing samples. Then, we tested the effect of age on mean Euclidean distance among tissues using the Spearman's correlation test. To study only the age effect on PC scores without the tissue effect, we performed the following: (1) we removed the tissue-specific effects from the data by scaling the expression levels of each gene to mean = 0 and sd = 1 in each tissue separately, (2) we combined the four scaled expression matrices, and (3) we conducted PCA on the combined dataset (*Figure 1—figure supplement 2*).

## Age-related gene expression change

To identify genes showing age-related expression change in each tissue, we used Spearman's correlation coefficient between individual age and expression level separately for development and ageing periods. To capture potential nonlinear but monotonic changes in expression, we chose the non-parametric two-sided Spearman's correlation test for both periods. We have used two-sided tests for all statistical tests throughout the article except the permutation tests. Significance of age-related genes was assessed with the FDR (FDR-corrected p-value<0.1 cutoff, calculated with the Benjamini–Hochberg [BH] procedure; *Benjamini and Hochberg, 1995*) using the 'p.adjust' function in the R base library. Throughout the article, BH procedure with 0.1 cutoff was used for multiple test corrections of all statistical tests.

## Functional associations

We tested the functional associations of age-related gene expression change in separate tissues for each period (development and ageing) separately, employing the gene set over-representation analysis (GORA) procedure with GO (*Ashburner et al., 2000*) BP categories using the 'topGO' package v.2.44 (*Alexa and Rahnenfuhrer, 2019*). We applied the 'classical' algorithm and performed Fisher's exact test on categories that satisfy the criteria of a minimum 10 and maximum 500 number of genes. We used the whole set of expressed genes (n = 15,063) as the background. p-Values were corrected for multiple testing using the BH procedure. Categories with FDR-corrected p-value<0.1 were considered as significant.

## Correlation between age-related gene expression changes in different tissues

We calculated Spearman's correlation coefficients between age-related gene expression change $\rho_{gene}$ values (i.e. correlation between gene expression levels and age) calculated per gene in each tissue pair (*Figure 1c*). In order to test the statistical significance of the correlations, we used a permutation scheme as the expression levels across tissues are not independent but belong to the same mice. In order to account for the dependence, the individual ages were permuted in each round, but the permuted values were kept constant across tissues (similar to permutation tests applied in *Dönertaş et al., 2017*; *Işıldak et al., 2020*; *Dönertaş et al., 2018*). Specifically, we performed 1000 permutation rounds. In each round, we randomised the individual ages using the 'sample' function in R, while keeping the permuted age labels constant for individuals across tissues. We calculated the age-related gene expression changes with permuted ages in development and ageing datasets separately, thus simulating the null distribution with no age effect in each period. We then calculated the Spearman's

correlation coefficient between the age-related expression levels from the permutations across tissues and assigned the p-value by calculating the proportion of permuted calculations with a more extreme correlation. All permutation tests in the article were performed as one-sided tests. The estimated false-positive proportion (eFPP; proportion of false positives among all true non-significant results (true negatives + false positives)) was calculated as the median value of expected values divided by the observed value (*Figure 1—source data 1*).

## Shared gene expression changes across tissues

We summarised the number of shared age-related genes among tissues for up- and downregulated genes separately using FDR-corrected p-value<0.1 (*Figure 1—figure supplement 5*). The development and ageing datasets were tested separately. For each gene, we counted the number of tissues with the same direction of expression change with age. We calculated this overlap statistic among tissues (1) using genes with FDR-corrected p-value<0.1 and (2) with all genes without using any significance cutoff (*Figure 1e*, *Figure 1—figure supplement 4*).

## Permutation test

We again used a permutation scheme to assess the significance of shared age-related genes to account for the dependence among tissues. We tested the significance of shared up- and downregulated genes, selected with or without an FDR cutoff, in development and in ageing periods separately. We used the age-related expression change values ($\rho'_{gene}$) calculated by permuting individual ages, 1000 times. To test the significance of the overlap of significantly up- or downregulated genes (FDR-corrected p-value<0.1) among tissues, we used the following procedure: (1) for each permutation round, we ranked the $\rho'_{gene}$ values for each tissue in each period separately. (2) We chose the highest $N_u$ (to test the upregulation) or lowest $N_d$ (to test the downregulation) number of genes, where $N_u$ and $N_d$ are the number of significantly up- or downregulated genes, respectively, in a given tissue (FDR-corrected p-value<0.1). (3) For each permutation round, we calculated the number of overlaps across tissues using the chosen gene sets, that is, the number of tissues with the same direction of expression changes with age for those genes. Doing this for 1000 permutation results yielded a null distribution representing the expected overlaps if there were no age effects. (4) We calculated the p-value as the proportion of 1000 permutations where the number of overlaps was higher than the observed value. The eFPP was calculated as the median number of overlaps in permutations divided by the observed value.

Likewise, to test the significance of the overlap of shared up- and downregulated genes selected without FDR cutoff, we used the same permutation scheme explained above, but this time using all the age-related expression changes created using permutations ($\rho'_{gene}$), without applying a significance cutoff for any tissue, and calculating the overlap across tissues in the same way.

## Functional associations

We tested the functional associations of shared expression change trends among tissues in each period separately following the GORA procedure using the same criteria and algorithms explained in the previous section. To test shared upregulated (n = 45) or downregulated genes (n = 138) in development, we chose all significant age-related genes across tissues (n = 10,305) in the development period as background. Since we could not identify any shared ageing-related genes across tissues (*Figure 1—figure supplement 5*), we did not perform a functional test for the ageing period.

## Analysis of gene expression reversals

We compared the direction of gene expression change during development and during ageing to identify reversal genes in each tissue separately. Genes showing upregulation (positive correlation with age) in development and downregulation (negative correlation with age) in ageing were assigned as up-down (UD) reversal genes, while the genes with the opposite trend (downregulation in development and upregulation in ageing) were assigned as down-up (DU) reversal genes. Without using any significance level for expression-age correlation values, we calculated the proportion of genes showing reversal by keeping the expression change direction in development the same, that is, UD% = UD/(UU + UD) and DU% = DU/(DD + DU).

## Permutation test

To test the significance of reversal proportions, we kept the developmental changes constant and randomly permuted the individual ages only in the ageing period (as described earlier). Among developmental upregulated genes, we calculated the UD% in each permutation, simulating a null distribution for UD reversal. We applied the same principle for the DU genes. Thus, we created a null distribution with the expected reversal ratios and tested the significance of observed values for each tissue separately (*Figure 1—figure supplement 8*).

## Functional associations

We used the GORA procedure as described earlier to test functional associations of reversal genes in each tissue but kept the developmental changes constant in the background. More specifically, we tested the functional enrichment of UD reversal genes against UU genes, and DU genes against DD genes. We thereby specifically test the functions associated with the reversal pattern, but not development-associated functions.

## Overlap of reversal genes: Permutation test

We tested the significance of overlap using the same permutation scheme described above. Specifically, among developmental up- (or down-) regulated genes shared among tissues, we constructed null distributions by calculating the ratio of UD vs. UD + UU (or DU vs. DU + DD) genes shared among tissues, identified in 1000 random permutations of individual ages only in the ageing period (*Figure 1—figure supplement 9*). The number of shared upregulated genes was $n_{up}$ = 2255 (one gene excluded since it has constant expression in one tissue in ageing period), and the number of shared downregulated genes was $n_{down}$ = 2209.

## Tissue convergence and divergence calculations using CoV

For each individual mouse, for each gene (n = 15,063), we calculated the inter-tissue CoV estimate using normalised expression levels from the four tissues, dividing the standard deviation by the mean. We studied inter-tissue expression-variation change with age in development and ageing periods separately using two approaches: (1) using the change in mean or median CoV across genes and (2) studying significant CoV patterns at the single-gene level.

### Mean/median CoV across all genes

We assessed transcriptome-wide variation among the tissues of each individual mouse by calculating the mean (or median) CoV of genes and then performing the Spearman's correlation test between mean-CoV (or median-CoV) and individual age.

### CoV at the single-gene level

In the second approach, we tested the correlation between the CoV value of a gene and individual age for each commonly expressed gene using the Spearman's correlation test. p-Values were corrected for multiple testing using the 'BH' procedure. We used FDR-corrected p-value<0.1 as cutoff. The genes showing positive correlation between CoV and age were called 'divergent,' and the ones showing negative correlation were called 'convergent' (*Figure 2b*). Genes that display a divergent pattern during development and convergent pattern in ageing (without using a significance level) were called divergent-convergent (DiCo) genes (n = 4802).

### Permutation test

To test the significance of DiCo genes (n = 4802), we kept the developmental divergent genes constant (n = 9058, without a significance cutoff) and randomly permuted the individual ages only in the ageing period (as described earlier). Among developmental divergent genes, we calculated the DiCo% for each permutation, simulating a null distribution for the DiCo pattern (*Figure 2—figure supplement 12*).

### Clustering of DiCo genes

We used the k-means algorithm to cluster DiCo genes according to their CoV or expression changes with age separately (*Figure 2—figure supplements 2–3*). To find the optimum number of clusters for

both procedures, we applied gap statistics using the 'clusGap' function in the 'cluster' package v.2.1.2 with 500 simulations (*Tibshirani et al., 2001*). We used the 'kmeans' function in base R with 'iter.max = 20' and 'nstart = 50' parameters to cluster CoV values or expression levels which were standardised to mean = 1 and sd = 0 across genes.

## Effect of gene expression trajectories on DiCo

To identify potential non-monotonic expression changes with age that could not be detected with the Spearman's correlation coefficient, we clustered all expressed genes (n = 15,063) in each tissue separately using the k-means algorithm following the same steps explained above (*Figure 1—figure supplements 12–15*). The list of genes belonging to each cluster is given in *Figure 2—source data 1*. Then, for each cluster, separately in each tissue, we performed a Fisher's exact test to assess if a particular cluster pattern is enriched or depleted in DiCo genes relative to all other expressed genes (the background).

## Functional association analysis

To test the functional associations of the genes showing the DiCo pattern among tissues, we performed GSEA using GO BPs. We retrieved developmental divergent genes (with $\rho_{\text{CoV-age}} > 0$, n = 9058) and multiplied these $\rho_{\text{CoV-age}}$ values with the ones calculated in the ageing period. Therefore, the genes with a negative value represent a DiCo pattern, while the ones with a positive value represent a DiDi pattern. We then ranked the genes according to the calculated product values and sought enrichment for the upper and lower tail of the distribution using the KS test implemented in the 'clusterProfiler' package v.4.0.0 (*Yu et al., 2012*). The 'gseGO' function was used with parameters: 'nPerm = 1000, minGSSize = 10, maxGSSize = 500 and pValueCutoff = 1'. Therefore, the enriched categories for the genes in the lower tail of the distribution would represent DiCo enrichment. Categories with FDR-corrected p-value<0.1 were considered as significant.

We summarised DiCo-enriched categories into representative ones following *Dönertaş et al., 2021* and used hierarchical clustering on gene similarities among categories. The tree was cut into 25 clusters. For each cluster, we chose as representative the category that has the highest mean Jaccard similarity to the other categories in the same cluster. Then, we calculated the mean age-expression correlation across all the genes in each representative category in each tissue and in each period. As the unrelated categories, those with the low within-cluster similarity were grouped into one cluster, we denoted them 'Other GO,' and performed the same clustering steps to further summarise them (*Figure 4—figure supplement 1*).

We further sought functional enrichment among DiCo genes that were clustered with the k-means algorithm for both CoV and expression clusters separately (*Figure 2—figure supplements 2–3*). Genes in each cluster were tested among all DiCo genes using the same GORA procedure as described before.

## Jackknife to test the Di/Co ratio between development and ageing

We tested the significance of divergent/convergent gene ratios using a jackknife resampling procedure in development and in ageing periods separately. Leaving out an individual in each iteration, we recalculated the number of significant divergent and convergent genes and their ratios. As we could not obtain any gene with significant CoV changes when the youngest adults were left out due to the decreased power, standard error and confidence interval calculation was not possible. Instead, we report the range of pseudovalues. We note that the range of ratios in leave-out samples do not contain the value 1 either in the development (0.41–0.49) or in the ageing (1.20–2.83) period (*Figure 2e*).

## Pairwise tissue DiCo test

In order to further verify the inter-tissue DiCo pattern that we observed between development and ageing periods, we used a different approach based on expression correlations among tissues. We calculated pairwise Spearman's correlation coefficients among tissues of the same individual mouse using all commonly expressed genes among the tissues (n = 15,063). For each tissue pair, we tested the correlation between age and inter-tissue expression correlations using the Spearman's correlation test in development and in ageing periods separately. In addition, we calculated the mean (or median)

of all six pairwise tissue correlations for each individual mouse and tested the correlation between age and average inter-tissue expression correlations using the Spearman's correlation test (*Figure 2— figure supplement 6*).

## Determination of tissue-specific genes

To identify which tissue(s) contribute to the reversal pattern, we assigned each gene to a tissue to identify tissue-specific expression patterns. First, we calculated an ES between the expression of a gene in a tissue versus other three tissues using the development samples only, and repeated this procedure for all tissues. Hence, we obtained ES for each commonly expressed gene in each tissue. ES was calculated using the 'Cohen's d' formula defined as the difference between the two means divided by the pooled standard deviation. We then assigned each gene to a tissue in which the gene has the highest ES. Finally, we retrieved only the fourth quartile (>Q3) of genes assigned to a tissue to define tissue-specific expression. Using this approach, we identified 3766 tissue-specific genes in total (cortex: 1175; lung: 839; liver: 986; muscle: 766 genes).

### Enrichment test with the direction of age-related change

We tested the association between tissue specificity and age-related expression change during ageing using Fisher's exact test. Specifically, we constructed a contingency table with two categorical variables; the first variable defines the direction (either positive or negative) of maximum expression change during ageing identified in a tissue-specific gene, which is determined by the slope of the regression between log2 age and expression. The second variable defines whether this maximum expression change identified in a tissue-specific gene occurs in its native tissue or not (either yes or no). Hence, a positive odds ratio (OR) suggests that (1) either the expression of genes decreases the most in their native tissue and/or (2) the expression of genes increase the most in a non-native tissue during ageing.

### Enrichment of tissue-specific genes in DiCo genes

We tested the association between tissue specificity (being either tissue-specific [n = 3766] or not [n = 11,297]) and the DiCo pattern (either showing DiCo [n = 4802] or not [n = 10,261]) using the Fisher's exact test, calculating the enrichment of tissue-specific genes within DiCo genes.

## Additional publicly available bulk tissue transcriptome datasets

### Jonker

We downloaded the raw data from the GEO database with GSE34378 accession number (*Jonker et al., 2013*) and followed the same analysis pipeline described above using all the samples from five tissues ('brain – cortex,' 'lung,' 'liver,' 'kidney,' 'spleen') of 18 female mice comprising 90 samples in total. This dataset represents the ageing period of the mouse, ranging from 90 to 900 days. Using the oligo package v.1.56.0 (*Carvalho and Irizarry, 2010*), we retrieved the expression matrices and performed 'rma' normalisation followed by removing the probesets that were annotated to more than one gene. We confined the analysis to only the protein-coding genes expressed in at least 25% of all samples. The resulting 17,661 genes were log2 transformed (after adding 1) and quantile normalised using the preprocessCore library (*Bolstad, 2020*) across all samples. Downstream analysis was the same as described above.

### Schaum

We downloaded the raw count matrix from the GEO database with GSE132040 accession number (*Schaum et al., 2020*) and performed the same filtrating steps as described above. We discarded the samples that have less than 4 million reads, which was the cutoff used in the article. We restricted the analysis to only protein-coding genes expressed in at least 25% of the samples that have expression in four tissues ('brain,' 'lung,' 'liver,' 'muscle'). One individual was removed from the analysis due to being an outlier in PCA after visual inspection (mouse ID: '3m7,' PCA plots before and after outlier removal are present in our GitHub repository; *hmtzg, 2022*). Final dataset contained 16,806 protein-coding genes from 37 mice that range from 3 to 27 months of age covering the ageing period. There were 11 female mice ranging from 3 to 21 months of age and 26 male mice ranging from 3 to 27 months of age. We performed the same normalisation method and downstream analyses described above. We

extended the analysis to eight tissues ('brain,' 'heart,' 'kidney,' 'liver,' 'lung,' 'muscle,' 'spleen,' 'subcutaneous fat') which were chosen based on the highest number of individuals that have the same tissue samples and that cover the whole ageing period (3–27 months). For the fat tissue, 'subcutaneous fat' was chosen as representative tissue which has the highest number of samples among all minor fat tissues. After performing the same preprocessing steps explained above, the final dataset contained 17,619 genes from 26 mice. Downstream analysis was the same as above.

## GTEx

We downloaded the processed GTEx v8 dataset (*Battle et al., 2017*) from the data portal and repeated the analysis in human tissues. We first confirmed our results in the same 4 tissues ('brain – cortex,' 'lung,' 'liver,' 'muscle – skeletal') and then expanded the analysis to 10 tissues ('adipose – subcutaneous,' 'artery – tibial,' 'brain – cerebellum,' 'lung,' 'muscle – skeletal,' 'nerve – tibial,' 'pituitary,' 'skin – sun exposed [lower leg],' 'thyroid,' 'whole blood'). In order to choose which tissues to analyse, we first chose the minor tissues with the highest number of samples for each major tissue, which prevents the representation of the same tissue multiple times. We then performed hierarchical clustering of tissues based on the presence of samples from the same individuals (*Figure 2—figure supplement 13*) and cut the tree into three clusters based on visual inspection. We selected the cluster with the highest number of overlapping individuals to analyse. The same procedure was followed for both 4- and 10-tissue analyses. In particular, we restricted the analysis to the individuals with samples in all tissues analysed and with a death circumstance of 1 (violent and fast deaths due to an accident) and 2 (fast death of natural causes) on the Hardy scale (n = 47 for 4 tissues, n = 35 for 10 tissues). We removed duplicated genes from the analysis. Similar to our analysis with the mice data, we used only the protein-coding genes that are expressed in at least 25% of all samples, totalling 16,197 for 4 tissues and 16,305 for 10 tissues. The TPM values obtained from the GTEx data portal were log2 transformed (after adding 1) and quantile normalised using the preprocessCore library (*Bolstad, 2020*) in R. Downstream analysis was the same as other datasets. To study the sex-specific convergence patterns, we repeated the same analysis separating female (n = 11) and male (n = 36) individuals.

## Comparison of datasets

We compared the age-related expression change patterns across tissues of all datasets analysed using Spearman's correlation coefficient. We used the 'pheatmap' function from pheatmap package v1.0.12 (*Raivo, 2019*) using hierarchical clustering (*Figure 4—figure supplement 2a*).

We performed Fisher's exact test to test the enrichment of convergent genes among datasets during ageing. We used only the convergent genes in ageing in our dataset (n = 7748) for comparison. For GTEx and Schaum et al. datasets, we performed enrichment for the same four tissues as our dataset and also for the larger sets, indicated as GTEx10 and Schaum8, respectively (*Figure 4—figure supplement 2b*).

## Regulatory analysis

We used MiRTarBase (downloaded on 03/08/2021; *Hsu et al., 2011*, *Hsu et al., 2014*) and TRANSFAC (downloaded on 03/08/2021; *Matys et al., 2003*; *Matys et al., 2006*) resources from the Ma'ayan lab database (*Rouillard et al., 2016*) for miRNA and transcription factor binding site (TFBS) enrichment analyses, respectively. As the database contains target information only for human HGNC IDs, we first converted those IDs to human Ensembl IDs and then to mouse Ensembl IDs only for the one-to-one ortholog genes using 'getBM' and 'getLDS' functions from the biomaRt package. In total, we analysed 235 miRNAs associated with 5458 target genes and 158 TFs associated with 7427 target genes. We conducted the overrepresentation analysis in the same way as for the DiCo functional enrichment analysis: specifically, we tested the targets of each regulator for enrichment in -Co genes (convergent genes in ageing) among Di- genes (divergent genes in development) used as background to keep developmental patterns fixed. We restricted the analysis for miRNA and TFs that have at least five target genes. After multiple testing correction with the BH procedure, we found no enrichment among either of the regulator types. Enrichment results are given in *Figure 4—source data 1*.

## Heteroscedasticity tests on the DiCo pattern

To test the hypothesis that the convergence pattern observed in the ageing period could be explained by the increased noise with age, thus regression towards the mean, we performed two distinct heteroscedasticity tests to compare DiCo genes against the lifelong-divergent genes (DiDi). In the first, we followed the method used to measure heteroscedasticity in *Işıldak et al., 2020* and *Kedlian et al., 2019*. We first fit a linear model between log2-transformed age and expression level for each gene in each tissue (*Kedlian et al., 2019*; *Işıldak et al., 2020*; *Somel et al., 2006*). This represents the variability of error along the explanatory variable, age. Then, we calculated Spearman's correlation coefficient between the absolute residual values and age, which can be used as an estimate of heterogeneity change with age. We compared the heterogeneity change values of DiCo and DiDi genes using a two-sided KS test in each tissue. In the second approach, we used the 'ncvTest' function from the 'car' package v.3.0.11 (*Fox and Weisberg, 2018*), which is a chi-squared test for heteroscedasticity estimated using a linear model. Again, we compared the heteroscedasticity measures of DiCo and DiDi genes using a two-sided KS test in each tissue.

## Single-cell RNA-seq

### Preprocessing

We used the Tabula Muris Senis dataset (*Schaum et al., 2020*) for scRNA-seq analysis as it is the only dataset to our knowledge that includes time-series samples covering old age and the tissues present in our dataset. Seurat-processed FACS data of the tissues lung, liver, skeletal muscle, and non-myeloid brain were downloaded from the figshare database (*Pisco, 2020*). The Seurat package v.4.0.0 (*Stuart et al., 2019*) was used to retrieve the expression matrix of the cells that are annotated to cell types in the original article. Each tissue contains samples from three time points: 90- (3 months), 540- (18 months), and 720-day-old (24 months) mice, totalling 14 samples each in lung, liver, and brain, and 9 samples in liver. We excluded cell types with less than 15 cells among all samples and excluded genes if the expression level is 0 for all cells at a given age. This resulted in a median number of 99–382 cells assigned to cell types, 6–24 cell types and 16,951–22,122 genes across tissues. Using 3-month-old mice, we calculated cell-type-specific expressions in each tissue. Specifically, we first calculated the mean expression levels among cells of an individual mouse for each cell type, and then calculated the mean among individuals to obtain an average expression value for each cell type. Uniprot gene symbols were converted to Ensembl gene IDs using the 'biomaRt' R package (*Durinck et al., 2009*).

### Deconvolution

We used cell-type-specific expression profiles of 3-month-old mice to estimate relative contributions of cell types to the transcriptome profiles of tissues in our mouse dataset. For a given tissue in our mouse dataset, we used single-cell expression profiles of that tissue from the Tabula Muris Senis dataset. We used a linear regression-based deconvolution method for each tissue using three genesets: all genes (n = [12,492, 12,849]), DiCo genes (n = [4007, 4106]), and non-DiCo genes (n = [8485, 8743]). Regression coefficients were used as relative contributions of cell types according to the following linear model:

$$Yi = a + b_{j1} * X_{i1} + b_{j2} * X_{i2} + ... + b_{jn} * X_{in},$$

where *i* represents the tissue, $Y_i$ is the expression level of a sample in a tissue, $b_{j1...jn}$ represents the relative contributions of the n cell types in a tissue, and $X_{i1...in}$ represents the expression levels of the n cell types in a tissue.

We then tested the effect of age on cell-type contributions ($b_{j1},...b_{jn}$) using the Spearman's correlation test in development and in ageing.

### Cell-type similarities and their change during ageing

To investigate the contribution of cell-autonomous changes to inter-tissue convergence in ageing, we calculated pairwise cell-type expression correlations among tissues and studied how these correlations change with age. Based on pairwise correlations in the 3-month age group, we identified the maximally and minimally correlated cell-type pairs among tissues. Specifically, for each cell type in a given tissue, we chose the minimally correlated cell type in each of the other three tissues. For example,

for each of the 10 cell types in the liver, we chose the minimally correlated cell type among the 15 cortex cell types, the minimally correlated cell type among the 24 lung cell types, and the minimally correlated cell type among the 6 muscle cell types. We repeated this procedure for all cell types in all four tissues, resulting in 54 cell-type pairs. Then, we calculated Spearman's correlation coefficients between age and minimally correlated cell-type pairs identified in the 3-month-age group. Likewise, we repeated the same analysis for the maximally correlated cell-type pairs among tissues.

### Permutation tests

To test whether DiCo genes are significantly more associated with cell-type proportion changes than non-DiCo genes, we performed a permutation test based on a resampling procedure. For each tissue, we took random samples among all genes (n = [12,492, 12,849]) with size $N$, where $N$ is the number of DiCo genes in that tissue, and repeated the deconvolution analysis as explained above. By calculating cell-type proportion changes with age for each random sample repeated 1000 times, we created the null distribution for each cell type. Then, we calculated the p-values as the number of random samples having the same or higher cell-type proportion change values divided by the observed value (cell-type proportion changes with DiCo genes).

We applied a similar permutation scheme as explained above to test cell-type similarity change differences between DiCo and non-DiCo genes. For each random sample of non-DiCo genes with size $N$, we calculated the pairwise correlations among cell types of tissues and identified maximally and minimally correlated cell types in the 3-month-age group. Then, we calculated age-related changes of those correlations using Spearman's correlation coefficient to construct the null distribution.

### Analysis of within-tissue convergence of cell types

Analogous to inter-tissue convergence analysis, we also studied intra-tissue convergence of cell types in scRNA-seq data by calculating CoV among cell types within a tissue for each individual of ages 3 months, 18 months, and 24 months, separately. We filtered the data to obtain cell types present in at least two individual mice in every time point for each tissue which yielded 4, 7, 20, and 6 cell types in brain, liver, lung, and muscle, respectively. We then tested the mean CoV (or CoV per gene) change with age using Spearman's correlation test.

## Acknowledgements

We thank Wolfgang Enard and Wulf Hevers for help with the mouse experiments and sharing samples, Nurcan Tuncbag, Nihal Terzi Çizmecioğlu, and the whole METU CompEvo team for helpful comments and fruitful discussions, and Zeliha Gözde Turan and Melih Yıldız for the critical reading of the manuscript and their suggestions. This work was supported by EMBL (HMD), the Scientific and Technological Research Council of Turkey (TÜBİTAK 2232, MS), the Science Academy (of Turkey) BAGEP Award (MS), and a METU Internal Grant (BAP, MS). The publication of this article was funded by the Open Access Fund of the Leibniz Association and the Leibniz Institute on Aging – Fritz Lipmann Institute (FLI), Jena, Germany. The FLI is a member of the Leibniz Association and is financially supported by the Federal Government of Germany and the State of Thuringia.

## Additional information

### Funding

| Funder | Grant reference number | Author |
| --- | --- | --- |
| European Molecular Biology Laboratory | | Handan Melike Dönertaş |
| Scientific and Technological Council of Turkey | 2232 | Mehmet Somel |
| Science Academy (Turkey) BAGEP Awards | | Mehmet Somel |

| Funder | Grant reference number | Author |
|---|---|---|
| METU Internal Grant | | Mehmet Somel |
| Leibniz Institute on Aging – Fritz Lipmann Institute (FLI) | Open Access Fund | Handan Melike Dönertaş |
| Leibniz Association | Open Access Fund | Handan Melike Dönertaş |

The funders had no role in study design, data collection and interpretation, or the decision to submit the work for publication.

## Author contributions

Hamit Izgi, Conceptualization, Data curation, Formal analysis, Methodology, Software, Visualization, Writing – original draft, Writing – review and editing; Dingding Han, Investigation, Resources, Writing – review and editing; Ulas Isildak, Data curation, Formal analysis, Validation, Visualization, Writing – review and editing; Shuyun Huang, Investigation, Writing – review and editing; Ece Kocabiyik, Data curation, Formal analysis, Writing – review and editing; Philipp Khaitovich, Conceptualization, Project administration, Resources, Supervision, Writing – review and editing; Mehmet Somel, Conceptualization, Funding acquisition, Methodology, Project administration, Resources, Supervision, Writing – original draft, Writing – review and editing; Handan Melike Dönertaş, Conceptualization, Data curation, Formal analysis, Funding acquisition, Methodology, Project administration, Supervision, Validation, Visualization, Writing – original draft, Writing – review and editing

## Author ORCIDs

Hamit Izgi http://orcid.org/0000-0002-4030-3132
Philipp Khaitovich http://orcid.org/0000-0002-4305-0054
Mehmet Somel http://orcid.org/0000-0002-3138-1307
Handan Melike Dönertaş http://orcid.org/0000-0002-9788-6535

## Ethics

Human subjects: Data involving human subjects were obtained from a published dataset, GTEx portal (https://www.gtexportal.org/home/datasets, with accession phs000424.v8.p2). Hence, no ethical statement is required.

Post-mortem samples were obtained from 16 C57BL/6J mice aged between 2 days and 904 days. All mouse experiments were overseen by the Institutional Animal Welfare Officer of the Max Planck Institute for Evolutionary Anthropology (MPI-EVA). They were performed according to the German Animal Welfare Legislation, ("Tierschutzgesetz") and registered with the Federal State Authority Landesdirektion Sachsen (No. 24-9162. 11-01 (T62/08)). The mice were sacrificed for reasons independent of this study, their tissues were harvested and frozen immediately, and stored at -80°C.

## Decision letter and Author response

Decision letter https://doi.org/10.7554/eLife.68048.sa1
Author response https://doi.org/10.7554/eLife.68048.sa2

# Additional files

## Supplementary files

• Supplementary file 1. Gene set over-representation analysis (GORA) of age-related genes in tissues. Tissue-specific age-related gene expression changes and functional enrichment test results, performed with GORA using 'topGO' package.

• Supplementary file 2. Gene set over-representation analysis (GORA) of shared age-related genes among tissues. Functional enrichment for shared genes across tissues. The same GORA that was performed for *Supplementary file 1* was used to test the enrichment of shared up-/downregulated genes in development among the background genes which are chosen as the all-significant age-related genes across tissues in development. We did not apply the test for the ageing period as there were no shared ageing-related expression changes.

• Supplementary file 3. Gene set over-representation analysis (GORA) of reversal patterns. Functional enrichment for gene expression reversals. GORA was performed with the same criteria as explained above. Up-down reversal genes were tested against up-up genes, and down-up reversal

genes were tested against down-down genes in each tissue.

• Supplementary file 4. Gene set over-representation analysis (GORA) of divergence-convergence (DiCo) gene clusters determined with coefficient of variation (CoV) values. Functional enrichment of DiCo genes clustered with k-means algorithm according to their CoV values. GORA was performed using gene sets in each cluster (*Figure 2—figure supplement 2*) which were tested among all DiCo genes.

• Supplementary file 5. Gene set over-representation analysis (GORA) of divergence-convergence (DiCo) gene clusters determined with expression levels. Functional enrichment of DiCo genes clustered with k-means algorithm according to their expression levels. GORA was performed using gene sets in each cluster (*Figure 2—figure supplement 3*) which are tested among all DiCo genes.

• Transparent reporting form

### Data availability

Sequencing data generated for this study have been deposited in GEO under accession code GSE167665. All data analysed during this study are included in the manuscript and supporting files. Source data files have been provided for all figures and figure supplements. Four additional and previously published datasets are used in this study: Jonker et al. 2013, GTEx Consortium et al. 2017, Schaum et al. 2020, and Tabula Muris Consortium 2020. All the code used to perform analyses is available in GitHub: https://github.com/hmtzg/geneexp_mouse (copy archived at swh:1:rev:1f2434f90404a79c87d545eca8723d99b123ac1c).

The following dataset was generated:

| Author(s) | Year | Dataset title | Dataset URL | Database and Identifier |
|---|---|---|---|---|
| Izgi H, Han D, Isildak U, Huang S, Kocabiyik E, Khaitovich P, Somel M, Donertas HM | 2021 | Bulk RNA-seq of mice covering the whole lifespan (2 days to 904 days) from four tissues | https://www.ncbi.nlm.nih.gov/geo/query/acc.cgi?acc=GSE167665 | NCBI Gene Expression Omnibus, GSE167665 |

The following previously published datasets were used:

| Author(s) | Year | Dataset title | Dataset URL | Database and Identifier |
|---|---|---|---|---|
| Jonker MJ, Melis JP, Kuiper RV, van der Hoeven TV, Robinson J, van der Horst GT, Breit TM, Vijg J, Dollé ME, Hoeijmakers JH, van Steeg H | 2013 | Aging Experiment | https://www.ncbi.nlm.nih.gov/geo/query/acc.cgi?acc=GSE34378 | NCBI Gene Expression Omnibus, GSE34378 |
| GTEx Consortium et al | 2017 | Gene TPMs | https://storage.googleapis.com/gtex_analysis_v8/rna_seq_data/GTEx_Analysis_2017-06-05_v8_RNASeQCv1.1.9_gene_tpm.gct.gz | GTEx Portal, phs000424.v8.p2 |
| Pisco A | 2020 | Official data release for Tabula Muris Senis | https://figshare.com/articles/dataset/Tabula_Muris_Senis_Data_Objects/12654728 | figshare, 10.6084/m9.figshare.12654728.v1 |
| Schaum et al | 2020 | Tabula Muris Senis: Bulk sequencing | https://www.ncbi.nlm.nih.gov/geo/query/acc.cgi?acc=GSE132040 | NCBI Gene Expression Omnibus, GSE132040 |

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
