## [Editor Report]

In this study, Izgi et al. investigated age-dependent gene expression pattern changes in male mice by analysing a new bulk RNA-seq data from four different tissues collected at different ages covering postnatal development and ageing. Gene expression patterns observed before sexual maturity show inter-tissue divergence, whereas convergence of gene expression profiles is observed after sexual maturity and during ageing, in a pattern that the authors call divergence-convergence or ‘DiCo.’ This observation may suggest that ageing results in at least a partial loss of tissue identity acquired developmentally.

---

## [Decision Letter]

**Decision letter after peer review:**

Thank you for submitting your article "Inter-tissue convergence of gene expression and loss of cellular identity during ageing" for consideration by *eLife*. Your article has been reviewed by 3 peer reviewers, one of whom is a member of our Board of Reviewing Editors, and the evaluation has been overseen by Matt Kaeberlein as the Senior Editor. The reviewers have opted to remain anonymous.

The reviewers have discussed their reviews with one another, and the Reviewing Editor has drafted this letter to help you prepare a revised submission.

Essential revisions:

1. The reviewers noted some potential issues with the normalization of the RNA-seq data that could affect the results, including the use of FPKM normalization, which may drive some of the observations due to lack of inter-sample-normalization (see specific comments by Reviewers #2 and 3). The authors would need to show that using appropriate count-based methods yields similar conclusions.

2. The authors often support their conclusions on low effect sizes and significance above usual thresholds. It would be important to explicitly discuss these as caveats, as well as strongly tone down these conclusions (including in the title.). More generally, the reviewers felt that potential alternative hypotheses needed to be clearly laid out, and potential ruled out, for the manuscript to stand more strongly (see specific comments by Reviewers #1, 2 and 3.)

3. In general, the reviewers felt that additional information on methods was required for reproducibility (see individual reviewer reports for specific points).

*Reviewer #1 (Recommendations for the authors):*

1. "Such "runaway development" phenomena may be due to insufficient natural selection pressure to optimise regulation after sexual reproduction starts." As stated, this sentence suggests that maladaptive traits that could lower an organism's total reproductive success would not be selected against as long as they only appear after the beginning of reproduction. I would perhaps ask that here the authors cite, not a general review of evolutionary theories of aging, but a publication that specifically proposes that alleles that are deleterious as early as the beginning of sexual reproduction nevertheless are not selected against.

2. Lines 84-91, 145-153, 229-240, 245-250, etc. Here the font size changes in the manuscript which I assume is not intended.

3. Figure 1 caption. "Similarities were calculated using Spearman correlations coefficient between expression-age correlations across tissues." Here I suspect that "Spearman correlations coefficient" was intended to be "Spearman's correlation coefficient".

*Reviewer #2 (Recommendations for the authors):*

Although the authors report an intriguing finding, there are major issues in the manuscript as it stands, notably concerning the clarity and rigor of the data analysis and manuscript.

1. For the analysis of their RNA-seq dataset across samples, the authors describe using the FPKM framework (methods, line 438). This is very problematic for a number of reasons – use of FPKM for differential gene analysis across samples is deprecated because it poorly controls for inter-sample variability [PMID: 22988256; PMID: 32284352]. FPKM are not comparable between samples because the normalization is sample-specific – thus, since no cross-library normalization step is performed, it should never be used for between sample comparison. Thanks to community benchmarking efforts, TMM or VST normalized counts are widely believed to be the superior choice for DGE analysis [PMID: 22988256; PMID: 32284352]. Thus, it is crucial that the authors correct their analysis to avoid the use of FPKM, and show that their conclusions would hold with an analysis on TMM or VST normalized counts.

2. There are issues regarding consistency in analytical choices throughout the paper.

a. Application of significance cutoffs vary widely in the manuscript when selecting specific gene sets. Reasoning should be provided for each case in which significance cutoff was (or not) applied (e.g. Line 111). For example, it is rather concerning that in Figure 1d, no gene was significantly up-regulated with age in the muscle (significance cutoff applied), but in Figure 1f, a large proportion of genes are shown to increase in expression with age in the muscle (significance cutoff not applied).

b. Authors are not consistent with the FDR threshold applied (e.g. FDR < 10% in Line 123, FDR 20% in Line 127). The reasoning for the chosen thresholds, as well as the different choices should be provided for cases in which higher than standard FDR is applied.

c. In Figure 1, authors show that the muscle presents with the least number of significantly changed gene expression patterns (up and down) with age. Additionally, in the manuscript, the authors explain that the lack of genes that show statistically significant change in the muscle is supported by a previous study that also showed "weak ageing transcriptome signature across multiple datasets (Line 137)." Thus, it is rather concerning that excluding the cortex, a tissue with a large number of differentially expressed genes, for CoV calculation resulted in greater significance.

d. The authors use a mix of multiple correction methods without clear explanations as to why a consistent measure isn't used (i.e. FDR/BH, BY) – BY test is justified for GO analysis line 472, but why BY was not used for DGE analysis is not clear (line 462).

e. A number of statistical analyses performed in the manuscript do not pass the standard significance thresholds (FDR < 5% or p < 0.05). The standard applied in the field is usually p-value or FDR < 5%, although relaxing this threshold can be done when dealing with noisier datasets. When using more relaxed FDR or p-value thresholds, the authors should also be careful to strongly tone down statements proportionally to the lower statistical support for their claims (eg. Line 240: "a SIGNIFICANT divergence-convergence pattern).

3. The authors use a dataset from exclusively male mice. Since aging is known to be extremely sex-dimorphic [PMID: 27304504], the authors need to explicitly discuss this caveat/limitation in the main text (i.e. these observations may not hold in female animals). If the authors wish to make a more general statement, a suggestion would be to reanalyze data from the recent Tabula Muris Senis Companion paper Schaum et al., [PMID: 32669715], which includes similar tissues, "developmental" time points (1 vs 3 months) and aging course (3 to 27 months), and both sexes to extend the findings. In addition, the Schaum dataset is also RNA-seq, and thus more directly comparable to the author's dataset compared to the microarray Jonker dataset used for partial validation (Figure 2 – supplement 8).

4. The authors should consider revising the title, since the current form seems to indicate that the work discovered s significant loss of cellular identity during aging. This statement is too strong based on the analyses of the paper. We suggest toning down the title, for instance to "Transcriptomic analyses suggest inter-tissue convergence of gene expression and loss of cellular identity during ageing".

*Reviewer #3 (Recommendations for the authors):*

Authors only analyze the gene sets that are correlated with age i.e. linearly go up or down with age. However, ageing trajectories for different genes can be very dynamic. This is the likely reason why only a few age-related changes were detected in some tissues. To assess the robustness of some of the key observations, authors should consider an orthogonal and unbiased clustering analysis similar to Figure 2 supplement to get the patterns for gene expression changes with age and to what extent DiCo is observed with different clusters.

Some of the key observations have either low effect size or are not statistically significant. For example, inter-tissue expression similarity during ageing and development (lines 115-119), the expression reversal (lines 158-161), changes in coefficient of variation with age (lines 202-207) etc. Authors clearly mention these in the manuscript, but it raises concerns on the extent to which some of these observations might be random, or due to idiosyncrasies of RNA-seq itself. These caveats should also be discussed.

In the global PCA based analysis for figure 1 and 2 combining both development and ageing data, the variance can be driven by one of the two states. For example, transcription divergence analysis in PCA in figure 1 based on development alone is not significant which suggests that the divergence of the developmental samples is driven in-fact by ageing. For PCA based analysis using separate PCAs for development and ageing will be more statistically sound.

Are there any known gene categories that have stronger expression reversal trends? For example, do known transcription factors or pathways that regulate development show DiCo if analyzed separately? Or is this trend only seen at the genome-wide level and completely stochastic?

For functional enrichment analyses authors only mention some selected functions in the text but GO terms have a lot of redundancy and it is unclear if there is any emerging functional trends looking beyond the redundancy or selected examples. It will be useful to thoroughly analyze and summarize the enriched GO terms to assess for any functional patterns that connect functional decline with ageing and convergent expression or DiCo.

Is there a significant overlap between mouse and human ageing related changes at the gene or pathway level?

Line 299 – linking the functional categories with cellular identity appears a bit hand-wavy. Stronger support would be needed to make this conclusion.

Some of the aspects need more methodological details:

For RNA-seq analysis, were all the samples normalized together, or were development and ageing samples normalized separately? It should be clarified in the methods.

Were the mouse perfused? The details should be added in methods. If the mouse were not perfused, a significant shared proportion of gene expression can be due to blood components. Also, the possibility that some of the shared changes could be due to infiltration of immune and other cell types in tissues should be mentioned.

More details for single cell deconvolution should be added in methods.

It is unclear to me how how the age related gene expression and PC correlation was performed. Was the loadings underlying PC axes used for correlation analysis? It should be elaborated in methods.

Some of the supplemental figures are missing.

---

## [Author Response]

Essential revisions:1. The reviewers noted some potential issues with the normalization of the RNA-seq data that could affect the results, including the use of FPKM normalization, which may drive some of the observations due to lack of inter-sample-normalization (see specific comments by Reviewers #2 and 3). The authors would need to show that using appropriate count-based methods yields similar conclusions.

We believe this comment mainly stems from the lack of clarity of our methods, which we improved in this version. We had used quantile normalisation on FPKM normalised values, which is an inter-sample-normalisation and normalises for overall library sizes between samples. In this revised version, we further repeat the analysis with VST normalisation following the DESeq2 pipeline and confirm the robustness of our results. The details are provided under the relevant comments below.

2. The authors often support their conclusions on low effect sizes and significance above usual thresholds. It would be important to explicitly discuss these as caveats, as well as strongly tone down these conclusions (including in the title.). More generally, the reviewers felt that potential alternative hypotheses needed to be clearly laid out, and potential ruled out, for the manuscript to stand more strongly (see specific comments by Reviewers #1, 2 and 3.)

We thank the reviewers for raising this issue. We have now (1) explained the difference between FPR (now called eFPP to prevent confusion) and FDR in the methods, to clarify that we use the same significance cutoff across different analyses, and also provide an additional measure that estimates the false positive proportion (not a p-value) based on permutations; (2) toned down the manuscript, including the title; (3) included additional discussion to summarise all the limitations laid out throughout the manuscript; (4) specifically tested the alternative hypothesis suggested by the reviewer #1; (5) proposed ageing-related increase in inter-cellular noise (heteroskedasticity) as a possible explanation for the convergence observed during ageing.

3. In general, the reviewers felt that additional information on methods was required for reproducibility (see individual reviewer reports for specific points).

We have re-written a significant proportion of the methods and included detailed explanations in the figure legends where applicable to improve reproducibility.

Reviewer #1 (Recommendations for the authors):1. "Such "runaway development" phenomena may be due to insufficient natural selection pressure to optimise regulation after sexual reproduction starts." As stated, this sentence suggests that maladaptive traits that could lower an organism's total reproductive success would not be selected against as long as they only appear after the beginning of reproduction. I would perhaps ask that here the authors cite, not a general review of evolutionary theories of aging, but a publication that specifically proposes that alleles that are deleterious as early as the beginning of sexual reproduction nevertheless are not selected against.

We thank the reviewer for the suggestion. The article by de Magalhaes and Church is the review article in which the authors revised the developmental theory of ageing especially in the light of genomic studies and thus is the original article where the ‘runaway development’ phenomenon was introduced in these exact terms. We now also included the original article by Mikhail V. Blagosklonny, where a related phenomenon, the hyperfunction theory was first proposed (Blagosklonny, 2006) – however, this is again a conceptual review where the theory was first proposed based on observations on many different grounds. Therefore, we have now also included papers with experimental support for this notion (Ezcurra et al., 2018; Lind et al., 2019). Moreover, we have now included a statement in the discussion to convey the message that we observe a reversal in the majority of the genes, but this does not necessarily negate the developmental or hyperfunction theory, which might be relevant for a limited number of pathways and may contribute to the ageing phenotype (lines 550-553). Finally, we now cite (Fisher, 1930; Medawar, 1952; Williams, 1957), which suggests the force of natural selection declines with age.

2. Lines 84-91, 145-153, 229-240, 245-250, etc. Here the font size changes in the manuscript which I assume is not intended.

Thank you for pointing this out – we have now corrected it.

3. Figure 1 caption. "Similarities were calculated using Spearman correlations coefficient between expression-age correlations across tissues." Here I suspect that "Spearman correlations coefficient" was intended to be "Spearman's correlation coefficient".

Thank you – we have changed it and carefully checked the manuscript one more time to prevent such typos.

Reviewer #2 (Recommendations for the authors):Although the authors report an intriguing finding, there are major issues in the manuscript as it stands, notably concerning the clarity and rigor of the data analysis and manuscript.1. For the analysis of their RNA-seq dataset across samples, the authors describe using the FPKM framework (methods, line 438). This is very problematic for a number of reasons – use of FPKM for differential gene analysis across samples is deprecated because it poorly controls for inter-sample variability [PMID: 22988256; PMID: 32284352]. FPKM are not comparable between samples because the normalization is sample-specific – thus, since no cross-library normalization step is performed, it should never be used for between sample comparison. Thanks to community benchmarking efforts, TMM or VST normalized counts are widely believed to be the superior choice for DGE analysis [PMID: 22988256; PMID: 32284352]. Thus, it is crucial that the authors correct their analysis to avoid the use of FPKM, and show that their conclusions would hold with an analysis on TMM or VST normalized counts.

We thank the reviewer for their comment. We had used FPKM followed by quantile normalisation, which performs an inter-sample normalisation. We had mentioned this at lines 443, 662, and 682 in the previous version, but had not explained this normalisation procedure in detail. Now the relevant section in the methods is updated to explain the quantile normalisation method (line 690). The main reason we choose quantile normalisation is that we analysed both RNA-seq and microarray data and this normalisation method can be applied to both, following other background normalisations and transformations (i.e. FPKM + log2 transformation for RNA-seq and RMA correction and log2 transformation for microarray). However, we agree with the reviewer that use of VST and TMM are more conventional as these are already implemented in the widely used analysis pipelines of DESeq2 and EdgeR.

We thus repeated the analysis of our dataset using VST and confirmed the main results.

We replicated our findings presented in Figure 1 using VST-normalised data. The results are presented in Figure 1—figure supplement 10 and 11:

i) We first compared VST- and QN-processed data in terms of age-related expression change trends in each tissue and period separately, without applying a significance cutoff (across n=14,973 genes). We found high correlations in all 8 comparisons (⍴_dev_=[0.81, 0.94], ⍴_ageing_=[0.83, 0.93], Figure 1—figure supplement 11).

ii) Repeating PCA analysis with VST-normalised data, we show that variation in gene expression (n=14,973 genes) is largely explained by tissue differences across PC1, 2, and 3 (ANOVA p<10^-16^ for PC1-3). In PC4, which explains 7% of the total variation (as opposed to 8% in Figure 1b, we observe an age-effect across tissues in development (Spearman’s correlation coefficient ⍴=[0.57, 0.99], nominal p<0.01 for each test except in muscle; Figure 1—figure supplement 10b). We also confirmed that tissues diverge during postnatal development, although the convergence during ageing was not significant (change in mean Euclidean distance among tissues with age in PC1-4 space, ⍴_dev_=0.937, p_dev_=0.00185; during ageing ⍴_ageing_=-0.58, p_ageing_=0.102, Figure 1-source data).

iii) Similar to Figure 1c results, we find that tissue pairs show weak correlations in their age-related expression change trends measured with VST-normalised data (n=14,973 genes), both during development (⍴=[0.19, 0.39]) and during ageing (⍴=[0.25, 0.34]). Furthermore, we find that the number of genes with the same direction of change (with or without significance cutoff) across four tissues with VST-normalised data is comparable to Figure 1d and Figure 1e result (Figure 1—figure supplement 10d-e).

iv) Reversal analysis with VST-normalised data revealed that ~50% (42-59%) of expressed genes (n=14,973) showed reversal patterns (without significance cutoff) in tissues comparable to Figure 1f result (Figure 1—figure supplement 10f). Comparing VST- and QN-normalised data, we found that 70-79% of the genes showed similar reversal or continuous patterns across tissues.

We confirmed our findings from Figure 2 with VST normalisation and presented the results in Figure 2—figure supplement 14:

i) We calculated CoV values across all expressed genes (n=14,973) among tissues in VST-normalised data and confirmed a significant mean CoV increase in development (Spearman’s correlation coefficient ⍴=0.99, p=10^-5^), consistent with QN normalised data that yielded ⍴=0.77 (p=0.041). During ageing, the decrease in mean CoV with age (⍴=-0.48, p=0.23) was comparable to the results shown in Figure 2a (⍴=-0.50, p=0.204).

ii) We replicated the difference between development and ageing in the proportion of genes showing divergent versus convergent patterns (at gene-wise FDR<0.1) with the VSTnormalised data (Figure 2—figure supplement 14d-e). Specifically, VST-normalised data displayed 89% divergence in development (among 3,476 significant genes) as opposed to 70% divergence in QN normalised data (among 2,581 significant genes). Likewise, during ageing, VST-normalised data displayed 68% convergence (among 19 significant genes) similar to 68% convergence in QN normalised data (among 62 significant genes).

We conclude that the choice of the normalisation method does not affect our results. We now present the VST-based results briefly in the text (lines 143-146, 263-265) to prevent redundancy and provide detailed explanations in the supplementary figure legends (Figure 1figure supplement 10, Figure 2—figure supplement 14). We now also explain the VST normalisation steps in the Methods (line 675).

2. There are issues regarding consistency in analytical choices throughout the paper.a. Application of significance cutoffs vary widely in the manuscript when selecting specific gene sets. Reasoning should be provided for each case in which significance cutoff was (or not) applied (e.g. Line 111). For example, it is rather concerning that in Figure 1d, no gene was significantly up-regulated with age in the muscle (significance cutoff applied), but in Figure 1f, a large proportion of genes are shown to increase in expression with age in the muscle (significance cutoff not applied).

We thank the reviewer for their comment. We actually analyse all the results in two ways: (1) using a consistent significance cutoff (multiple testing (FDR) adjusted p-value<0.1, or if no multiple testing was applied, p-value<0.05), and (2) without using a significance cutoff to assess transcriptome-wide trends. We emphasise this as soon as the first sentence of the Results section titled “Tissues involve common gene expression changes with age”. We now added a second sentence to emphasise this point:

“We next characterised age-related changes in gene expression shared across tissues by (i) studying overall trends at the whole transcriptome level and testing their consistency using permutation tests, and (ii) studying statistically significant changes at the single gene level.”

This has two motivations. First, given the low signal/noise ratios during mammalian ageing, identifying statistically significant patterns in individual genes is frequently difficult, but transcriptome-wide patterns can be more effectively identified. Second, observing the same trends using both approaches increases the robustness of our conclusions and may suggest that the observations are a genome-wide phenomenon, not restricted to a small number of genes. However, since transcriptome-wide trends are more prone to technical artefacts, we further test the genes that show statistically significant changes.

We now attempted to further clarify where a significance cutoff was and was not used in the analyses, by updating the text to emphasise the gene set of interest for each analysis (lines 131, 140).

We also thank the reviewer for pointing out the ambiguity about muscle genes. In Figure 1f, we study global reversal trends in each tissue for all the genes without any significance cutoff, i.e. only analysing expression change trends (following (Dönertaş et al., 2017)). Thus, although there are no statistically significant ageing-related genes in muscle tissue (Figure 1d), we observe a transcriptome-wide reversal trend (Figure 1f). We now updated the figure text to explain clearly what is presented (lines 183-187).

b. Authors are not consistent with the FDR threshold applied (e.g. FDR < 10% in Line 123, FDR 20% in Line 127). The reasoning for the chosen thresholds, as well as the different choices should be provided for cases in which higher than standard FDR is applied.

We thank the reviewer for pointing out that the difference between FDR (e.g. line 123 in the previous version) and FPR (e.g. line 127 in the previous version) was not clearly explained in the text. FDR is the multiple testing correction ‘false discovery rate’ applied using the ‘p.adjust’ function with BH procedure in R, which we did not clearly state in the first submission. We updated the text to explain that FDR is applied to correct the p-values for all statistical tests (functional association tests were corrected with BY procedure in the previous version but for consistency, we used FDR in all the tests in the current version – see point d.) (lines 711-714).

We have also updated the text to replace all “FDR<0.1” with “FDR corrected p-value<0.1”.

We had used the term FPR as an estimate of the false-positive rate calculated using random permutations (FPR = median number of positives observed across permutations / number of positives observed in the data). We provide this metric with permutation test results as additional information on the effect size, not as an alternative to FDR corrected p-values. However, thanks to the reviewer’s comment we realised that the phrase FPR could be confused with other metrics. We, therefore, redefined it as eFPP, ‘estimated false-positive proportion’ (lines 740-743). We now updated the relevant sections to make sure that these two definitions are not interchangeable. We also removed eFPP from the main text to avoid confusion and only provide this information only in the supplementary figures.

c. In Figure 1, authors show that the muscle presents with the least number of significantly changed gene expression patterns (up and down) with age. Additionally, in the manuscript, the authors explain that the lack of genes that show statistically significant change in the muscle is supported by a previous study that also showed "weak ageing transcriptome signature across multiple datasets (Line 137)." Thus, it is rather concerning that excluding the cortex, a tissue with a large number of differentially expressed genes, for CoV calculation resulted in greater significance.

We believe that the reviewer’s concern perhaps stems from our poor wording of CoV calculation in the text. For each individual, we calculated CoV across tissues using all expressed genes (n=15,063) regardless of their specific age-related expression changes (significant or not) in any tissue. Then, we summarised these gene-wise CoV values by calculating their mean to assess a genome-wide variation score among tissues of an individual and how these scores change with age (Figure 2a). We have now updated the relevant Results section to explain the CoV calculation in more detail (lines 221-223). We believe that the significant age-related expression changes in one tissue should not have a direct impact on the results as the number of significantly changed genes during ageing in the cortex was also very small (n=68).

Despite this, we repeated the same analysis, this time excluding the muscle tissue, which resulted in a nonsignificant but similar pattern during ageing (Spearman’s correlation test between mean CoV and age (n=15): ⍴_dev_=0.85, p_dev_=0.016; ⍴_ageing_=-0.29, p_ageing_=0.49, Author response image 1 -right panel), and the lack of significance again might be driven by the lack of one cortex sample.

The exclusion of the cortex data was motivated by the fact that we were missing an aged individual in this tissue. However, estimating CoV using only three points may be suboptimal, and we, therefore, decided to be conservative and thus removed this analysis (i.e. results excluding cortex) from the manuscript.

**Author response image 1. sa2fig1:** Age-related change in CoV summarised across genes excluding cortex or muscle.

Change in CoV using 15 samples from three tissues (one oldest individual (904 days of age) excluded due to its missing data in the cortex), either excluding cortex (left column) or muscle (right column) tissue. Each point represents the (a) mean CoV, (b) median CoV value of all protein-coding genes (15,063) for each mouse. x-axis is in log2 scale. The dashed grey line shows the start of the ageing period. The Spearman’s correlation coefficients and p values for each period are indicated separately on the plots.

d. The authors use a mix of multiple correction methods without clear explanations as to why a consistent measure isn't used (i.e. FDR/BH, BY) – BY test is justified for GO analysis line 472, but why BY was not used for DGE analysis is not clear (line 462).

We thank the reviewer for this point. For consistency, we now use FDR (BH) for multiple testing correction for all statistical tests throughout the text. We now updated the relevant sections (lines 711, 713-714).

e. A number of statistical analyses performed in the manuscript do not pass the standard significance thresholds (FDR < 5% or p < 0.05). The standard applied in the field is usually p-value or FDR < 5%, although relaxing this threshold can be done when dealing with noisier datasets. When using more relaxed FDR or p-value thresholds, the authors should also be careful to strongly tone down statements proportionally to the lower statistical support for their claims (eg. Line 240: "a SIGNIFICANT divergence-convergence pattern).

We thank the reviewer for the comment. We believe that the 10% FDR cutoff allows a compromise between overall false positives and false negatives, and is commonly used as a threshold in the transcriptomics literature, including recent analyses on the association between gene expression and splicing (DOI: 10.7554/*eLife*.67077), expression and neurodegeneration (DOI: 10.7554/*eLife*.64564), or expression and psychosocial experiences (DOI: 10.7554/*eLife*.63852).

Using a more stringent threshold would reduce the false positive probability per gene. However, our emphasis in this manuscript is not individual genes but transcriptome-wide patterns, which we test using the whole transcriptome without significance threshold, and also with significant genes (with an FDR corrected p-value<0.1). The particular instance where we use ‘significant’ in line 240 refers to the result of Spearman’s correlation test between mean pairwise correlations among tissues and age in development and ageing periods.

That said, we did follow the reviewer’s suggestion and toned down our conclusions. Accordingly, we changed the referred line to ‘we observed the same divergence-convergence pattern’ (lines 286-287).

3. The authors use a dataset from exclusively male mice. Since aging is known to be extremely sex-dimorphic [PMID: 27304504], the authors need to explicitly discuss this caveat/limitation in the main text (i.e. these observations may not hold in female animals). If the authors wish to make a more general statement, a suggestion would be to reanalyze data from the recent Tabula Muris Senis Companion paper Schaum et al., [PMID: 32669715], which includes similar tissues, "developmental" time points (1 vs 3 months) and aging course (3 to 27 months), and both sexes to extend the findings. In addition, the Schaum dataset is also RNA-seq, and thus more directly comparable to the author's dataset compared to the microarray Jonker dataset used for partial validation (Figure 2 – supplement 8).

We thank the reviewer for this valuable comment. In fact, we identify convergent trends in ageing in both female and male animals, in our dataset (only males) and in the Jonker et al., dataset (only females), respectively. We further find that convergent genes overlap between the two datasets, although modestly (OR = 1.22, p = 1.67x10^-8^, Figure 4—figure supplement 2b). This suggests that DiCo is not unique to a single-sex in mice. We now explain this in lines 430-434, 614-616.

We would also have been interested in testing possible sex dimorphism in DiCo, but this is not possible given technical differences between the two datasets (i.e. Jonker et al.,'s and ours).

The GTEx dataset includes both male and female humans and displays patterns consistent with convergence during ageing, although not significantly (Figure 2—figure supplement 811). Following the reviewer's point, we repeated the GTEx dataset analysis separately for females and males. We observed a stronger convergence in females than in males, although none of these results were significant (⍴_female_=-0.58, p_female_=0.059; ⍴_male_= -0.052, p_male_=0.77) (Figure 2—figure supplement 16). The female and male GTEx samples differ both in sample size (n=11 vs n=36, respectively) and age range (no male individuals within 20-29 and 70-79 age groups), which could explain the observed differences between sexes.

We now discuss possible sex-dimorphism in Discussion as a limitation of this study and suggest a more comprehensive analysis with different datasets in the future (line 612-620).

Finally, we thank the reviewer for suggesting including the Schaum et al., dataset. We now included this dataset, analysing both the same 4 tissues we have and a broader set of 8 tissues (Figure 2—figure supplement 17-20). However, we could not include the development period as the sample size was very small for the two time points; 1 and 3 months of age. We could not confirm the transcriptome-wide trend towards convergence during ageing in this dataset and found that the muscle and subcutaneous fat tissues showed the most divergent patterns.

However, we confirmed a strong association between the loss of identity and convergence during ageing. Not having the developmental period for the external datasets limited the complete replication of our original study, which focuses on the DiCo pattern, divergence followed by convergence. Nevertheless, a trend towards convergence during ageing was observed at the transcriptome level in Jonker et al., and GTEx and the association between convergence and loss of identity was evident across all datasets. We now updated the text to reflect that the support for the transcriptome-wide trend is weak and our most robust result is the association between identity loss and convergence.

We now add Figure 4—figure supplement 2 and Table 1 to summarise all results across datasets.

4. The authors should consider revising the title, since the current form seems to indicate that the work discovered s significant loss of cellular identity during aging. This statement is too strong based on the analyses of the paper. We suggest toning down the title, for instance to "Transcriptomic analyses suggest inter-tissue convergence of gene expression and loss of cellular identity during ageing".

Thank you for the suggestion – we have now toned down the title: ‘Inter-tissue convergence of gene expression during ageing suggests age-related loss of tissue and cellular identity’

Reviewer #3 (Recommendations for the authors):Authors only analyze the gene sets that are correlated with age i.e. linearly go up or down with age. However, ageing trajectories for different genes can be very dynamic. This is the likely reason why only a few age-related changes were detected in some tissues. To assess the robustness of some of the key observations, authors should consider an orthogonal and unbiased clustering analysis similar to Figure 2 supplement to get the patterns for gene expression changes with age and to what extent DiCo is observed with different clusters.

We thank the reviewer for the comment. The reason we used Spearman’s correlation was to capture potential non-linear but monotonic changes. However, we agree with the reviewer that Spearman’s correlation cannot identify non-monotonic changes. As suggested, we now performed k-means clustering for each tissue to identify age-related expression patterns and then performed an enrichment test to assess if DiCo genes are enriched or depleted in any cluster. The results are now presented as supplementary figures (Figure 1—figure supplement 12-15). Notably, we did not observe a bias towards a cluster having a linear pattern with age or not in any of the tissues. For example, the cortex tissue displayed 15 clusters of expression changes, among which 5 are enriched and another 5 are depleted in DiCo (Figure 1—figure supplement 12). Two of the DiCo enriched clusters (2/5) do not have linear trajectories with age (clusters 12 and 14) whereas three of the DiCo depleted clusters (3/5) display linear patterns (clusters 6, 7 and 15). We have now presented this result in the relevant section (lines 276-277).

Some of the key observations have either low effect size or are not statistically significant. For example, inter-tissue expression similarity during ageing and development (lines 115-119), the expression reversal (lines 158-161), changes in coefficient of variation with age (lines 202-207) etc. Authors clearly mention these in the manuscript, but it raises concerns on the extent to which some of these observations might be random, or due to idiosyncrasies of RNA-seq itself. These caveats should also be discussed.

We now added a limitations paragraph in the Discussion, detailing both this aspect and other limitations raised by the other reviewers (lines 605-620).

In the global PCA based analysis for figure 1 and 2 combining both development and ageing data, the variance can be driven by one of the two states. For example, transcription divergence analysis in PCA in figure 1 based on development alone is not significant which suggests that the divergence of the developmental samples is driven in-fact by ageing. For PCA based analysis using separate PCAs for development and ageing will be more statistically sound.

We thank the reviewer for this suggestion. We now followed the reviewer’s suggestion and analysed age-related convergence using PCA performed only on ageing samples. In this way, we now analyse Euclidean distance in PC1-4 (i) using all samples together (across the lifetime – Figure 1a), (ii) using samples from developmental period only (Figure 1—figure supplement 3a-b), and (iii) using samples from ageing period only (Figure 1—figure supplement 3d-e). For the PCA analysis based on all the samples together, change in mean Euclidean distance with age among tissues was significant both during development and ageing (⍴_dev_=0.99, p_dev_=1.5x10^-5^; during ageing ⍴_age_=-0.87, p_age_=0.0026). When we performed the PCA analysis separately for the two periods, we still observed a significant divergence among tissues during development (⍴=0.95, p=0.0008) and observed a marginally significant convergence during ageing (⍴=-0.64, p=0.0596). We hope that this analysis addresses the reviewer’s point. We also note that our earlier analysis had an error (we had used PC3-4 spaces to calculate the Euclidean distance for the mouse dataset, mistakenly excluding PC1-2), which we now corrected (Figure 1, Figure 1—figure supplement 3).

Are there any known gene categories that have stronger expression reversal trends? For example, do known transcription factors or pathways that regulate development show DiCo if analyzed separately? Or is this trend only seen at the genome-wide level and completely stochastic?

We thank the reviewer for the suggestion. We now used the MiRTarBase database for miRNAs and TRANSFAC for transcription factors and tested if any specific regulator is associated with DiCo pattern genes. We did not find any significant association (lines 353357). This may be consistent with DiCo being a transcriptome-wide and diffuse phenomenon caused by stochastic changes, as the reviewer indicates. It is also possible that not specific regulators but a change in their cooperative action influences the pattern, or we lack sufficient power with the current datasets to identify subtle effects of individual regulators. We now mention this also in the Discussion (line 601-603).

For functional enrichment analyses authors only mention some selected functions in the text but GO terms have a lot of redundancy and it is unclear if there is any emerging functional trends looking beyond the redundancy or selected examples. It will be useful to thoroughly analyze and summarize the enriched GO terms to assess for any functional patterns that connect functional decline with ageing and convergent expression or DiCo.

In the previous version, enriched GO terms were summarised according to their Jaccard similarities using the ‘emapplot’ function of the ‘enrichplot’ package. However, we agree with the reviewer that the enrichment network did not remove redundancy among the categories, and might have hidden unique and interesting functions.

We have now used another approach to summarise and interpret DiCo enriched categories in more detail. First, we clustered the categories based on the number of genes they share using hierarchical clustering and cut the tree into 25 clusters. For each cluster, we chose a representative category that has the highest mean-Jaccard similarity to the other categories in the same cluster (Dönertaş et al., 2021). Then, we calculated the mean expression changes (⍴ between expression and age) of the genes in the representative categories for each tissue, separately (Figure 4h). We believe that this new analysis provides better insight to our main findings such that tissue-related functions gained during development display opposing expression trends during ageing, i.e. reversal pattern, providing support to loss of tissuespecific expression and thus contributing to convergence during ageing. We have now updated the main text and methods (lines 343-351, 874-881).

Is there a significant overlap between mouse and human ageing related changes at the gene or pathway level?

We thank the reviewer for the suggestion. We now compare the correlations between ageing related expression changes and overlap between convergent genes during ageing (Figure 4 figure supplement 2). While mouse datasets, especially our data and Jonker et al., showed higher correlations among each other, the correlations between age-related expression changes in the human and mouse datasets were smaller but overall positive when we consider the same tissues in both datasets (average correlations between GTEx and mouse datasets are ⍴_Jonker_=0.12, ⍴_Our_dat_=0.13, ⍴_Schaum_=0.01). Convergent genes in ageing also showed weak but significant overlaps across datasets. Human dataset showed significant overlap with Jonker et al., and Schaum et al., datasets but not with ours. Nevertheless, the overlap for these datasets was also small. Importantly, apart from the differences in species, sex, age distribution and the technology used to generate the data, the small overlap between datasets may stem from the fact that the external datasets lack developmental samples. Thus, we can only compare convergence during ageing but not the DiCo pattern, the main focus of our study. Since only 62% of the convergent genes during ageing are divergent in development in our dataset, we should emphasise that low overlap for convergence does not rule out overlap across DiCo genes. Similarly, we use divergent genes during development as the background for our functional enrichment tests to find functional associations of DiCo. Since we lack the developmental background we performed functional enrichment for only the convergent genes but did not find any significant association for the GTEx dataset.

Line 299 – linking the functional categories with cellular identity appears a bit hand-wavy. Stronger support would be needed to make this conclusion.

Thank you for the suggestion. We have now toned down that particular line (347-350), and also revised the manuscript, in general, to make sure we do not overstate our findings.

Some of the aspects need more methodological details:For RNA-seq analysis, were all the samples normalized together, or were development and ageing samples normalized separately? It should be clarified in the methods.

We thank the reviewer for pointing out this ambiguity. We have combined all the samples together regardless of their age (development and ageing together) or tissue (n=63) and performed quantile normalisation on log2 transformed (after adding 1) FPKM values. As FPKM normalisation does not account for inter-sample variability, we adopted this approach to control for cross-library variability between different tissues and ages. We have rewritten the relevant section in the methods to explain the normalisation in more detail (line 666-685). The same approach is used for the VST-normalisation in this version and development and ageing periods are normalised together.

Were the mouse perfused? The details should be added in methods. If the mouse were not perfused, a significant shared proportion of gene expression can be due to blood components. Also, the possibility that some of the shared changes could be due to infiltration of immune and other cell types in tissues should be mentioned.

We thank the reviewer for this question. Indeed, infiltration of other cell types is a serious consideration in most bulk tissue analyses. Since we observe a similar trend between different cell types using scRNAseq data and that DiCo is particularly associated with tissue-specific genes and not with any bood or immune-related signature, we believe the effect should be minimal in our analysis. However, the new dataset we included in the analysis, Schaum et al., did use perfused mice and although the association between the loss of tissue identity and convergence was particularly strong, the transcriptome-wide trend did not support convergence. We now included the potential influence of infiltration of other cell types as a limitation of our study in the Discussion (line 606-612).

More details for single cell deconvolution should be added in methods.

We have now updated the deconvolution section in the methods to explain how we conducted the analysis in more detail (lines 1054-1064).

It is unclear to me how how the age related gene expression and PC correlation was performed. Was the loadings underlying PC axes used for correlation analysis? It should be elaborated in methods.

We performed PCA on individuals’ gene expression values and used these PC scores of the individuals (not the loadings) and their ages to perform the correlation analysis. We have now updated the relevant section in the methods (lines 693-696).

Some of the supplemental figures are missing.

We are sorry for this confusion, but having checked our submission we could not detect any missing files. In case this might be an issue with the submission portal, for the second round we will update the bioRxiv version of the article where the reviewers can check all files separately if needed.